# ASPSCR1::TFE3 orchestrates the angiogenic program of alveolar soft part sarcoma

Miwa Tanaka [1,2,3] ✉, Surachada Chuaychob [4], Mizuki Homme[1,5], Yukari Yamazaki[1,2], Ruyin Lyu[4], Kyoko Yamashita [6], Keisuke Ae[7], Seiichi Matsumoto[7], Kohei Kumegawa [3], Reo Maruyama [3], Wei Qu[8], Yohei Miyagi [9], Ryuji Yokokawa[4] & Takuro Nakamura [1,2] ✉

Alveolar soft part sarcoma (ASPS) is a soft part malignancy affecting adolescents and young adults. ASPS is characterized by a highly integrated vascular network, and its high metastatic potential indicates the importance of ASPS's prominent angiogenic activity. Here, we find that the expression of *ASPSCR1::TFE3*, the fusion transcription factor causatively associated with ASPS, is dispensable for in vitro tumor maintenance; however, it is required for in vivo tumor development via angiogenesis. ASPSCR1::TFE3 is frequently associated with super-enhancers (SEs) upon its DNA binding, and the loss of its expression induces SE-distribution dynamic modification related to genes belonging to the angiogenesis pathway. Using epigenomic CRISPR/dCas9 screening, we identify *Pdgfb, Rab27a, Sytl2,* and *Vwf* as critical targets associated with reduced enhancer activities due to the ASPSCR1::TFE3 loss. Upregulation of Rab27a and Sytl2 promotes angiogenic factor-trafficking to facilitate ASPS vascular network construction. ASPSCR1::TFE3 thus orchestrates higher ordered angiogenesis via modulating the SE activity.

Tumor angiogenesis is one of the most important processes in malignant progression and distant metastasis of cancer[1]. Migration, extension, and sprouting of endothelial cells are key mechanisms that initiate angiogenesis to construct vascular tubes that are required for oxygen and nutrition transport to cancer cells as well as cancer-cell intravasation[2]. However, this cancer-induced neovascularization frequently induces the construction of fragile blood vessels that exhibit abnormal distributions and reduced pericyte numbers, resulting in frequent vessel destruction and obstruction and insufficient blood supply in common cancers[3,4]. In contrast, there is a group of cancers that induce abundant and highly integrated blood vessels with abundant pericyte wrapping, and the presence of the pericytes and their derivatives affects the malignant potential of cancer. The group includes cancers originating from blood vessel-rich organs, such as the kidney, liver, and endocrine organs[5–7]. These cancers further demonstrate distinct metastatic phenotypes, maintaining epithelioid structures unrelated to epithelial mesenchymal transition[7,8]. The pericyte that wraps around the vascular tube plays a key role in angiogenic initiation and vessel stability in both neoplastic and non-neoplastic conditions[4,9]. Platelet-derived growth factor β (PDGFB), which is provided by both cancer cells and non-cancerous stromal cells, is important for pericyte recruitment, and the interaction between PDGFB and PDGF receptor β (PDGFRB) on pericytes is responsible for mature-blood-vessel stabilization[10,11].

[1]Division of Carcinogenesis, The Cancer Institute, Japanese Foundation for Cancer Research, Tokyo, Japan. [2]Department of Experimental Pathology, Institute of Medical Science, Tokyo Medical University, Tokyo, Japan. [3]Project for Cancer Epigenomics, The Cancer Institute, Japanese Foundation for Cancer Research, Tokyo, Japan. [4]Department of Micro Engineering, Graduate School of Engineering, Kyoto University, Kyoto, Japan. [5]Division of Cell Biology, The Cancer Institute, Japanese Foundation for Cancer Research, Tokyo, Japan. [6]Division of Pathology, The Cancer Institute, Japanese Foundation for Cancer Research, Tokyo, Japan. [7]Department of Orthopedic Oncology, Cancer Institute Hospital, Japanese Foundation for Cancer Research, Tokyo, Japan. [8]Department of Computational Biology and Medical Sciences, Graduate School of Frontier Sciences, The University of Tokyo, Kashiwa, Japan. [9]Molecular Pathology and Genetics Division, Kanagawa Cancer Center Research Institute, Yokohama, Japan. ✉e-mail: miwa.tanaka@jfcr.or.jp; takuron@tokyo-med.ac.jp

Alveolar soft part sarcoma (ASPS) is a slow-growing malignant neoplasm affecting adolescents and young adults[12,13]. Despite its less aggressive nature in the primary site, ASPS is a highly metastatic tumor, and patients exhibit poor outcomes due to the limitations of the current therapies[14–16]. ASPS is characterized by a blood vessel-rich alveolar structure with a highly integrated vascular network, which is responsible for frequent metastasis[13,17]. Blood-vessel abundance in ASPS suggests that the vascular network is an ideal target for effective therapies; however, vascular endothelial growth factor receptor (VEGFR)-inhibitor administration has not provided a cure for ASPS[14,16,18]. These findings indicate that there is an unknown ASPS angiogenic mechanism that should be clarified to facilitate enhanced therapy.

We established a mouse model for human ASPS[17] that effectively recapitulates ASPS phenotypes, such as alveolar structure with fine capillary network, frequent intra- and extravasation of tumor nests, and distant metastasis. Importantly, tumor-blood-vessel analysis in the mouse model and human ASPS clarified that these vessels are well encapsulated with PDGFRB- and smooth muscle actin-positive pericytes, and that tumor-cell enclosure by pericytes is observed in the blood stream during distant metastasis. The study revealed the unique angiogenic process in ASPS and underscored the importance of angiogenesis in ASPS development and progression.

ASPS is invariably associated with *ASPSCR1* and *TFE3* gene fusion due to t(X;17) chromosomal translocation[19]. *ASPSCR1::TFE3* encodes an oncogenic transcription factor using TFE3's basic helix-loop-helix (bHLH) domain as a DNA-binding domain[20,21]. Like other sarcoma-associated fusion genes, *ASPSCR1::TFE3* is essential for transforming target cells to induce ASPS in vivo[17,22]. As an aberrant MiT/TFE3 family transcription factor, ASPSCR1::TFE3 regulates genes involved in lysosomes, autophagy, and angiogenesis[17,21,23,24]. ASPSCR1::TFE3 also upregulates genes important for angiogenesis and tumor metastasis, such as *PDGFB*, *ANGPTL2*, and *GPNMB*[17,24,25], suggesting that ASPSCR1::TFE3 plays a central role in the angiogenesis characteristic of ASPS. In addition, its function in rather restricted cells-of-origin, such as embryonic mesenchymal cells, suggests that a proper epigenetic environment is important for the oncogenic and angiogenic function of ASPSCR1::TFE3.

Despite the strong circumstantial evidence for the association between ASPSCR1::TFE3 and angiogenesis, the mechanism by which ASPSCR1::TFE3 orchestrates angiogenesis in association with the proper epigenetic conditions remains unclear. Previous studies on oncogenic fusion transcription factors, such as EWSR1::FLI1 in Ewing sarcoma, have indicated that the functions of these transcription factors frequently interact with pre-existing and/or novel super-enhancers (SEs)[26].

In this study, we investigate the epigenetic landscape of mouse and human ASPS as well as genome wide DNA binding of ASPSCR1::TFE3. Interestingly, the expression of ASPSC::TFE3 is dispensable for tumor-cell maintenance in vitro; however, its expression is required for in vivo tumorigenesis and angiogenesis, and *ASPSCR1::TFE3*-expression loss induces drastic modifications in SE distribution. Therefore, to understand how ASPSCR1::TFE3 induces angiogenesis, we investigate the target SEs and downstream genes involved in angiogenesis using our ASPS model and an innovative organ-on-a-chip system.

## Results

### ASPSCR1::TFE3 is dispensable for cell growth and survival of ASPS cells in vitro but required for in vivo tumorigenesis

In our previous study, *ASPSCR1::TFE3* expression in murine embryonic mesenchymal cells could effectively induce highly metastatic tumors with human ASPS phenotypes when they were transplanted into nude and Balb/c mice[17]. These sarcoma cells proliferate well in vitro and are serially transplantable. However, we observed frequent *ASPSCR1::TFE3*-

expression loss or decrease in 21 of 32 cell lines during in vitro passage (Fig. 1, a–c). There were no significant differences in cell proliferation or morphology (Fig. 1d, Supplementary Fig. 1a, c). When *ASPSCR1::TFE3* was knocked down in the human ASPS-KY or ASPS1 cell lines, the in vitro growth potential was not significantly affected (Fig. 1d, Supplementary Fig. 1c). Moreover, acute loss of *ASPSCR1::TFE3* expression by siRNA-mediated knockdown in murine ASPS17 and ASPS25 cells did not affect cellular growth significantly (Supplementary Fig. 1d). Although the cause of the frequent *ASPSCR1::TFE3*-expression loss has not been fully clarified, we detected a loss of the 5' long terminal repeat promoter/enhancer of the retroviral vector used for *ASPSCR1::TFE3* transduction in at least one pair of murine ASPS cell lines, ASPS17 and ASPS17 null (null) (Supplementary Fig. 1b). These results indicate that *ASPSCR1::TFE3* expression is dispensable for in vitro ASPS-cell growth once they are transformed. In contrast, no tumor growth was observed when ASPS null cells were transplanted into nude mice (Fig. 1e). Histological examination of early tumorigenic lesions 4 and 14 days after transplantation revealed that FLAG-positive ASPS-cell foci were accompanied by Cd31-positive endothelial cells and Pdgfrb-positive pericytes in the ASPS17 transplanted areas, whereas no FLAG-positive cells and very few Cd31-positive and Pdgfrb-positive blood vessel components were observed in the subcutaneous part of ASPS null transplanted recipients (Fig. 1f, Supplementary Fig. 1e). Gene expression profiling revealed significant upregulation and downregulation of genes according to the *ASPSCR1::TFE3* status. We found 2,572 upregulated genes and 3,308 downregulated genes (1.5-fold threshold) in ASPS17 cells compared with those in ASPS null cells as well as 1583 downregulated and 2,102 upregulated genes (1.2-fold threshold) in ASPS-KY cells by knockdown of *ASPSCR1::TFE3* (Supplementary Data 1–4). Gene set enrichment analysis (GSEA) revealed enrichment of angiogenesis-related pathways, such as the VEGF, PDGF, and intracellular vesicle/granule-related pathways (Fig. 1g, Supplementary Fig. 1f, g). Similar enrichment results were obtained in the *ASPSCR1::TFE3* knockdown experiment using the human ASPS-KY and ASPS1 cell lines (Fig. 1h, Supplementary Fig. 1h, i). Downregulated expression of individual genes in these pathways such as *Rab27a*, *Sytl2*, *Pdgfb*, *Vwf*, and *Gpnmb* was confirmed by quantitative reverse transcription polymerase chain reaction (qRT-PCR) (Fig. 1g, h, Supplementary Fig. 1f, g, h, i). Common genetic pathways were observed according to the *ASPSCR1::TFE3* expression status among mouse and human ASPS cell lines (Supplementary Fig. 1j). Collectively, these results suggest that ASPSCR1::TFE3 regulates downstream target genes involved in angiogenesis to induce ASPS's blood-vessel-rich alveolar structure and in vivo tumorigenesis.

### ASPSCR1::TFE3 affects chromatin remodeling and activates its downstream target genes

ASPSCR1::TFE3 binds DNA using TFE3-derived basic helix-loop-helix (bHLH) and leucin zipper domains and modulates downstream-target-gene expression[21]. Drastic modulation of gene expression profiles in the absence of ASPSCR1::TFE3 suggests that chromatin activity may be affected. ChIP-seq analysis demonstrated that ASPSCR1::TFE3 binding peaks were predominantly associated with distal regions (69–78%) and 16–22% of the peaks were located at gene promoters both in mouse ASPS17 and human ASPS-KY cells (Fig. 2a). The Genomic Regions Enrichment of Annotations Tool (GREAT) pathway analysis for these ASPSCR1::TFE3 binding peaks identified genes involved in lysosomes, cytoplasmic vesicles, or vacuolar membranes, whose expression was upregulated in ASPS (Supplementary Fig. 2a). The consensus binding motif for the MiT/TFE family transcription factors, CACGTGAC, was most highly enriched in mouse and human ASPS (Fig. 2b). Previous studies have suggested that target-gene upregulation by ASPSCR1::TFE3 plays an important role in the ASPS development[17,21,23]. We subsequently examined the association between ASPSCR1::TFE3 and histone H3K27ac-binding signals in both mouse and human ASPS.

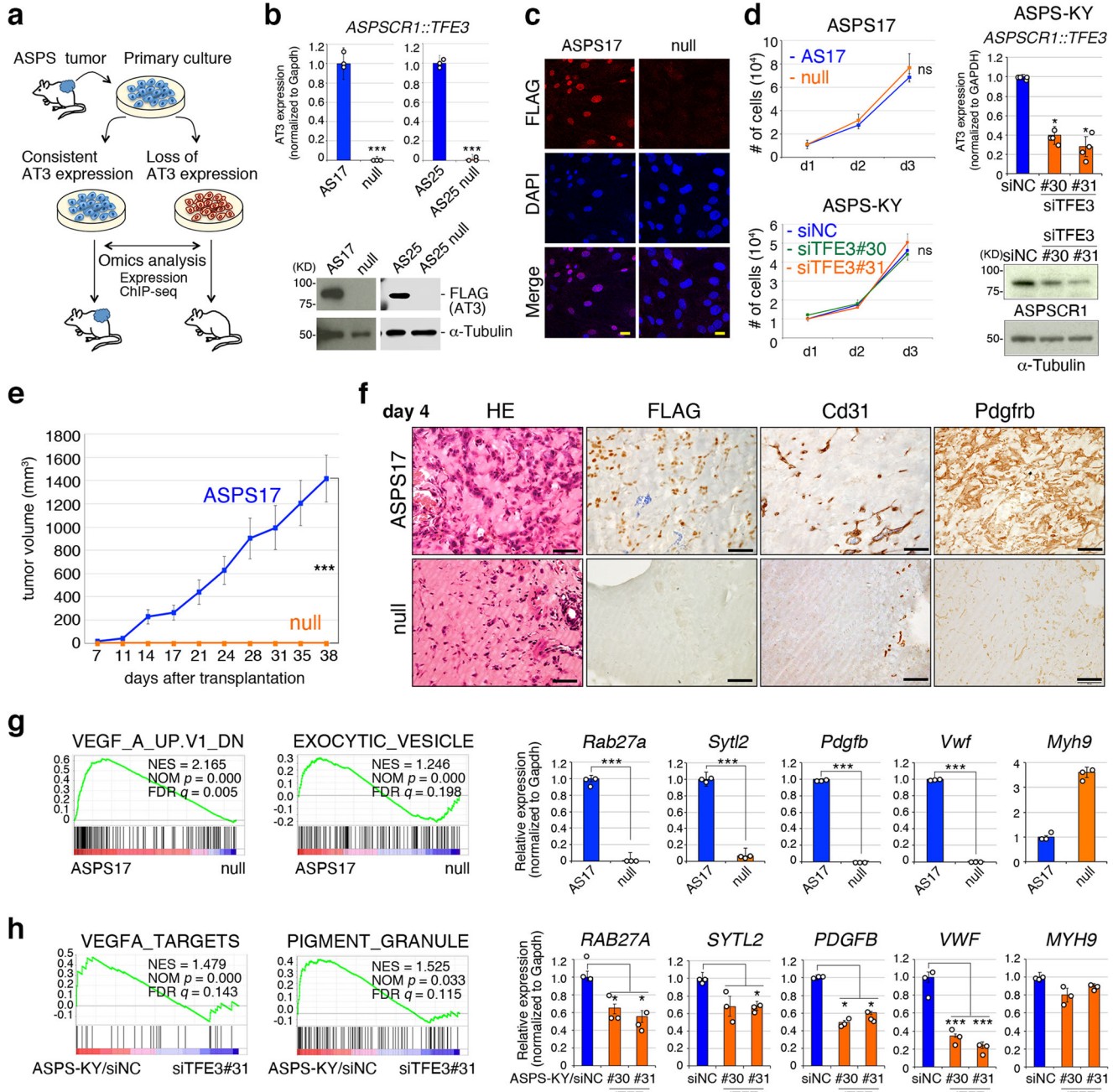

**Fig. 1 | *ASPSCR1::TFE3* expression is dispensable for cell maintenance in vitro but required for in vivo tumor development in ASPS. a** Schematic illustration of functional analysis in ASPS cells with or without *ASPSCR1::TFE3* expression. **b** ASPSCR1::TFE3 expression in ASPS17 and ASPS25 was compared with their null counterparts at the transcriptional level (top) (*n* = 3 per group) and protein levels (bottom, two representative immunoblots from eight independent experiments). **c** Loss of the ASPSCR1::TFE3 protein in ASPS null cells exhibited by immuno-fluorescence using the anti-FLAG antibody. Scale bar: 20 μm. Experiments represent five biological replicates. **d** Cell proliferation of mouse ASPS17 and ASPS null cells (left, top), and human ASPS-KY cells with siRNA treatment (left, bottom) (*n* = 4 per group). ns: no significance. The efficiency of *ASPSCR1::TFE3* knockdown is shown at transcriptional (*n* = 4 per group) and protein levels (right, representative immunoblots from five independent experiments). **e** Suppression of tumor development by loss of *ASPSCR1::TFE3* expression. Average tumor volumes with SD are shown in the recipient transplanted with ASPS17 and ASPS null cells (*n* = 6 mice/ 12 independent tumors per group). The tumor volume was measured using 2 tumors per mouse. **f** Histology of transplanted area with ASPS17 and ASPS null cells 4 days after transplantation. Hematoxylin and eosin (HE) staining and immuno-histochemistry with indicated antibodies. Scale bar: 50 μm. Experiments represent three biological replicates. **g** Gene set enrichment analysis (GSEA) showing enrichment of VEGF and exocytic vesicle pathways between ASPS17 and ASPS null cells. Normalized enrichment scores (NES), nominal *p*-values, and FDR *q*-values are indicated (left). The *p*-value is computed through the two-sided permutation test (*n* = 1000 randomizations) adjusted the Benjamini-Hochberg procedure. Quantitative RT-PCR (qRT-PCR) showing downregulation of *Rab27a, Sytl2, Pdgfb*, and *Vwf* in ASPS null cells while *Myh9* expression was increased (*n* = 3 per group). **h** GSEA showing enrichment of VEGFA and pigment granule pathways by comparing human ASPS-KY cells with and without knockdown of ASPSCR1::TFE3 (left). Downregulation of *RAB27A, SYTL2, PDGFB*, and *VWF* is shown (*n* = 3 per group). Statistical analyses in (**b, d, e, g, h**) were performed by two-sided Student's *t*-test and *marks adjusted *p*-value < 0.05, **marks adjusted *p*-value < 0.01 and ***marks adjusted *p*-value < 0.001. The data presented as mean ± SD. Source data are provided as a Source Data file.

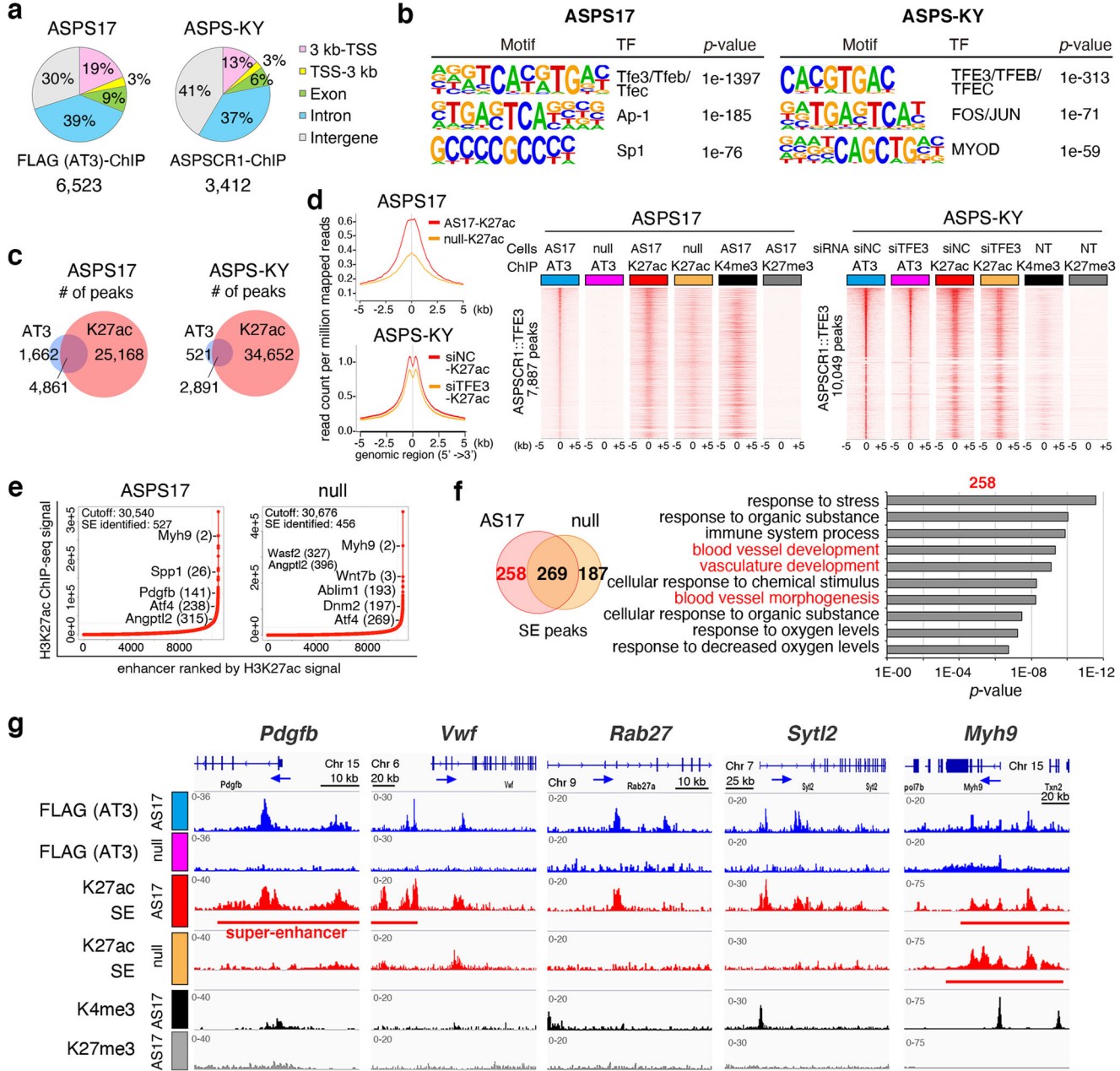

**Fig. 2 | The loss of *ASPSCR1::TFE3* expression modulates distribution of super-enhancers (SEs). a** The genomic distribution of ASPSCR1::TFE3 6523 and 3412 bound sites in mouse and human ASPS, respectively, showing predominant binding in distal regions. Common bound sites in two biological replicates were analyzed. **b** De novo motif enrichment analysis of ASPSCR1::TFE3 binding regions in mouse (left) and human (right) ASPS. The top three motifs identified are shown. Motif enrichment and *p*-value was calculated using Fischer's exact test. **c** Venn diagrams showing the number of common binding regions between ASPSCR1::TFE3 and H3K27ac in mouse and human ASPS. **d** Composite plots showing a significant reduction in H3K27ac signals in the absence of ASPSCR1::TFE3 in both mouse and human ASPS (left). Heat maps showing ASPSCR1::TFE3, H3K27ac, H3K4me3, and H3K27me3 signals in murine ASPS17 and human ASPS-KY cells. Reduction in H3K27ac signals is observed in both cells (right). **e** Enhancers are ranked by increasing H3K27ac signals in ASPS17 and ASPS null cells. Using the ROSE algorithm, 527 and 456 enhancers were defined as SEs in ASPS17 and null cells, respectively. **f** Venn diagram showing overlapping and distinct SEs (left). Enrichment of Gene Ontology biological process for 258 ASPS17-specific SEs, showing inclusion of angiogenesis pathways in red (right). *p*-value was calculated using a binominal test. **g** ChIP-seq track at *Pdgfb, Vwf, Rab27a, Sytl2,* and *Myh9* genomic loci, showing the association between ASPSCR1::TFE3 and H3K27ac binding in ASPS17 cells. Significant loss of H3K27ac signals and/or SEs in ASPS null cells are exhibited. The *Myh9* genomic locus is shown as an example of SEs unaffected by *ASPSCR1::TFE3* loss.

A frequent association between ASPSCR1::TFE3 and H3K27ac binding was observed (75% and 85% of ASPSCR1::TFE3 binding in mice and humans, respectively) (Fig. 2c, d). H3K27ac-binding signals around the ASPSCR1::TFE3 binding regions were significantly reduced by loss of or diminished *ASPSCR1::TFE3* expression (Fig. 2d, Supplementary Fig. 2b, c) while the total number of H3K27ac-binding peaks was not reduced (Supplementary Fig. 2b), suggesting that ASPSCR1::TFE3 potentially

modulates global histone modifications and chromatin activation. Thereafter, we compared SE profiles between ASPS17 and ASPS null cells and found that 49% of SEs were lost by loss of *ASPSCR1::TFE3* expression (Fig. 2e, f, Supplementary Data 5–7). Pathway analysis of 258 SEs that disappeared after *ASPSCR1::TFE3* loss revealed blood vessel-related pathway enrichment (Fig. 2f, g, Supplementary Fig. 2d). Significant differences in SE distribution were also evident in human

ASPS cells after *TFE3* knockdown (Supplemental Fig. 2e). Pathway analysis of ASPSCR1::TFE3-positive SEs revealed overlapping blood vessel activity-associated pathway enrichment (Supplementary Fig. 2f, g). These results strongly suggest that ASPSCR1::TFE3 partially regulates angiogenesis-associated target genes by modulating SE.

## The BET protein inhibitor JQ1 suppresses angiogenesis-associated SEs and inhibits ASPS in vivo growth

SEs are cis-regulatory elements targeted by the core regulatory circuitry and mediator complex[27]. BET-domain proteins play an important role in the SE functions, and the inhibition of their activities renders them effective targets for the SE-associated malignant potential of cancer[27,28]. In vivo treatment of murine and human ASPS cells transplanted into nude mice with JQ1, a BET-domain inhibitor, significant suppressed tumor growth (Fig. 3a, Supplementary Fig. 3a). This growth suppression was accompanied by remarkable angiogenic inhibition, exhibiting a reduction in CD31-positive endothelial cells and PDGFRB- or alpha-smooth muscle actin (aSMA)-positive pericytes (Fig. 3b, Supplementary Fig. 3b). Gene expression analysis of JQ1-treated ASPS cells in vitro using GSEA and Ingenuity Pathway Analysis (IPA) demonstrated PDGF-associated and angiogenesis-related pathway enrichment (Fig. 3c, d, Supplementary Fig. 3c). Brd4 is a major JQ1 target, and ChIP-seq analysis revealed a significant reduction in Brd4 binding around the ASPSCR1::TFE3 binding peaks in mouse and human ASPS (Fig. 3e, Supplementary Fig. 3d). A total of 23,924 out of 34,800 Brd4-binding peaks disappeared after JQ1 treatment, and the GREAT analysis of these peaks revealed angiogenesis-related pathway enrichment (Fig. 3f). Moreover, the involvement of JQ1 suppressed Brd4 binding, and the angiogenesis pathway was affected in an ASPSCR1::TFE3-dependent manner (Supplementary Fig. 3e). In addition, Brd4-binding signals within SEs were significantly decreased by JQ1 treatment (Supplementary Fig. 3f, g). ChIP-seq analysis also demonstrated that Brd4-binding signals were reduced in SEs and active enhancers associated with ASPSCR1::TFE3 target genes, namely, *Rab27a, Sytl2, Pdgfb*, and *Vwf*, whose expression was downregulated by *ASPSCR1::TFE3* loss (Fig. 3g, h, Supplementary Fig. 3h, i). In contrast, there was also a subset of genes, including *Myh9* and *Neat1* whose enhancers were not affected by the JQ1 treatment. Although it is reported that JQ1 treatment suppresses tumor growth through downregulation of *c*-Myc expression[29], c-Myc expression remained unchanged in mouse and human ASPS cells (Supplementary Fig. S3j). The CDK7/8 inhibitor, THZ1, which also targets SEs in certain cancers[30], did not exhibit growth suppressive effects on ASPS or angiogenesis-related gene downregulation (Supplementary Fig. 3k, l), suggesting that chromatin modifications on angiogenesis-associated SEs by ASPSCR1::TFE3 may be effective in a Brd4-specific manner. Overall, JQ1 treatment targets SEs associated with ASPSCR1::TFE3-related angiogenic function, and the results underscore the usefulness of epigenetic ASPS therapies.

## Epigenomic CRISPR screening identifies ASPSCR1::TFE3 target genes involved in angiogenesis and in vivo tumor development

SE modification by ASPSCR1::TFE3 and JQ1 treatment strongly suggests that downstream target genes regulated by these SEs play a critical role in ASPS-related angiogenesis, and that the identification of these targets will inform future ASPS therapies. Therefore, epigenetic CRISPR screening with dCas9-KRAB was performed to identify ASPSCR1::TFE3 targets that are required for angiogenesis and in vivo tumor growth. A total of 494 SEs and active enhancers in which H3K27ac signals were reduced by >40% due to *ASPSCR1::TFE3* loss were selected, and 7716 gRNAs (average 12.8 per enhancer) were designed to target these enhancers (Fig. 4a, Supplementary Data 8). To suppress target-enhancer activities, ASPS17 cells were introduced with lentiviral vectors bearing gRNAs and dCas9-KRAB, and pools of lentivirus-transduced cells were transplanted into nude mice (Supplementary

Fig. 4a). Locus-specific repression effects were confirmed by introducing individual lentiviruses (Supplementary Fig. 4b). We applied Model-based Analysis of Genome-wide CRISPR/Cas9 Knockout (MAGeCK) algorithm[31] to identify 45 loci with significantly depleted gRNAs in tumors (Fig. 4b, c, Supplementary Fig. 4c, Supplementary Data 9 and 10). Topologically associated domains (TADs) were defined by Hi-C analysis (Fig. 4d), and 271 candidate target genes included in the same TAD as each SE were selected. Forty-four candidate genes were further selected by downregulated expression in ASPS null cells and 25 genes were found upregulated in human ASPS samples comparing with other types of sarcoma (Fig. 4e, Supplementary Fig. 4i, Supplementary Data 11). Six candidate genes, namely, *Ccbe1, Pdgfb, Rab27a, Syngr1, Sytl2*, and *Vwf*, were finally selected by functional annotations and subjected to an in vivo validation assay. After effective knockout of each gene was validated (Supplementary Fig. 4d, e), the tumor clones were transplanted into nude mice, and tumor growth was monitored daily. Despite comparable proliferation rates in vitro (Supplementary Fig. 4f), significant suppression of in vivo tumor development was observed by *Pdgfb, Rab27a, Sytl2*, and *Vwf* knockout (Fig. 4f). As shown by *ASPSCR1::TFE3*-expression loss and JQ1 treatment, tumor-growth suppression by each knockout cell was accompanied by significant angiogenic inhibition (Supplementary Fig. 4g). The vascular structure of ASPS is characterized by a highly integrated vessel system comprising pericytes and endothelial cells. The present epigenetic screening identified Sytl2 and Rab27a, both of which are involved in intracellular trafficking of cytoplasmic vesicles[32], as well as angiogenic factors Pdgfb and Vwf, indicating that the ASPSCR1::TFE3-modulated chromatin activity orchestrates the core angiogenesis program of ASPS.

To evaluate the expression status of ASPSCR1::TFE3 target genes identified in the present study, their expression was examined using clinical bone and soft part sarcoma samples and human sarcoma cell lines. Increased *RAB27A, SYTL2*, and *VWF* expression was observed in human ASPS among the six sarcoma types (Fig. 4g). *PDGFB* expression in ASPS was the second highest following dermatofibrosarcoma protuberance, in which *PDGFB* fuses with *COL1A1* and is highly upregulated[33]. Upregulated *RAB27A, SYTL2*, and *VWF* expression was confirmed in two human ASPS cell lines, ASPS-KY and ASPS1, in comparison with two Ewing sarcomas, two synovial sarcomas, and one osteosarcoma cell line (Supplementary Fig. 4h). No significant differences were observed in *PDGFB* expression among ASPS, Ewing, and synovial sarcoma cells. Upregulation of *RAB27A, SYTL2, PDGFB*, and *VWF* in human ASPS was observed when expression of these four genes was compared among five sarcoma types using microarray data deposited to Gene Expression Omnibus (https://www.ncbi.nlm.nih.gov/geo/) (Supplementary Fig. 4i). Immunohistological examination revealed diffuse cytoplasmic RAB27A and SYTL2 staining, in accordance with their upregulated expression (Fig. 4h). VWF was also positive with focally strong staining. The mechanism of uneven VWF accumulation remains unclear, and the clarification of its relationship with endothelial sprouting is rather intriguing.

## Intracellular trafficking of angiogenic factors is promoted by the Rab27a/Sytl2 axis

As described previously, the small GTPase protein Rab27a binds to cytoplasmic vesicles and facilitates their trafficking to the plasma membrane in collaboration with Sytl/Slp family proteins[32]. To clarify the role of the Rab27a/Sytl2 axis in ASPS, Pdgfb and Gpnmb secretion was examined, since Gpnmb is important for extravasation of tumor cells[17]. *Rab27a* or *Sytl2* knockouts significantly suppressed Pdgfb and Gpnmb secretion (Fig. 5a). A co-immunoprecipitation experiment showed interaction between Rab27a and Sytl2, and the Rab27a W73G mutant, in which the binding activity for Sytl2 was lost[34], failed to exhibit increased secretion activity (Fig. 5a, Supplementary Fig. 5a, b). *Rab27a, Sytl2*, or *Pdgfb* knockout and

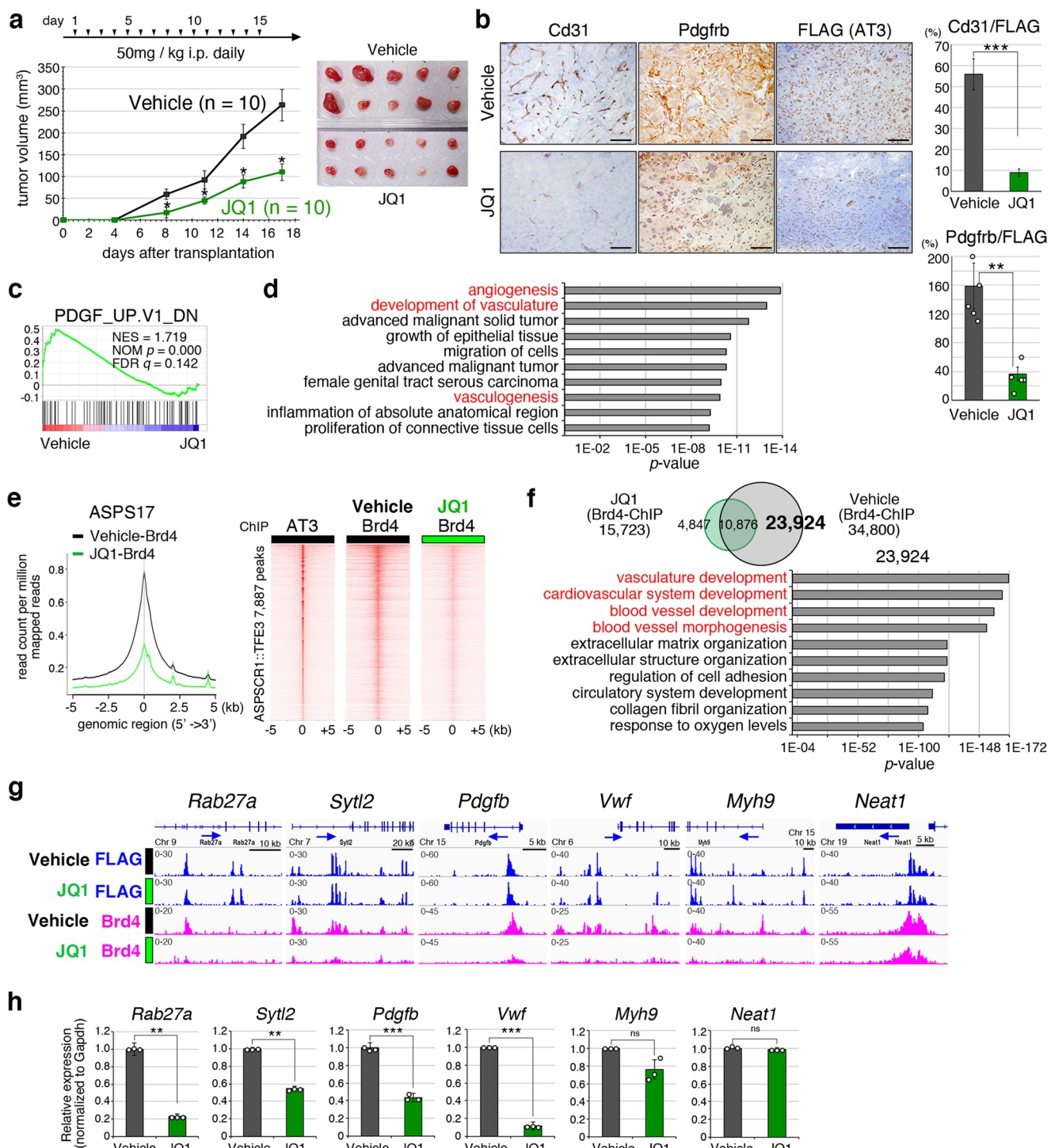

*ASPSCR1::TFE3* loss resulted in decreased pericyte growth induction when these cells' conditioned media were used (Fig. 5b), suggesting that the secretion of growth factors such as Pdgfb may be induced by ASPSCR1::TFE3 via upregulation of downstream target genes. Thereafter, Pdgfb and Gpnmb intracytoplasmic trafficking was examined in ASPS17, *Rab27a* knockout, *Sytl2* knockout, and ASPS null cells. Fluorescence recovery after photobleaching (FRAP) significantly delayed Pdgfb, Gpnmb, Angptl2 in *Rab27a* knockout, *Sytl2* knockout, and ASPS null cell membrane trafficking (Fig. 5c, d, Supplementary Fig. 5c). *Rab27a* re-expression rescued Pdgfb and Gpnmb trafficking, whereas the W73G mutant did not (Supplementary Fig. 5d). Likewise, von Willebrand factor (VWF) secretion was also significantly reduced by *Rab27a* and *Sytl2* knockout and by

*ASPSCR1::TFE3*-expression loss (Fig. 5e). Although its large molecular weight precluded the FRAP assay, the intra-cytoplasmic distribution of VWF indicated that VWF is predominantly localized in the perinuclear region in ASPSCR1::TFE3-, Rab27a-, or Sytl2-reduced conditions (Fig. 5f, Supplementary Fig. 5e). Nexinhib 20, a small-molecule inhibitor for the interaction between Rab27a and Sytl1 (Supplementary Fig. 5f), a close Sytl2 homolog[35], suppressed Pdgfb, Gpnmb, and VWF secretion and Pdgfb, Gpnmb, and Angptl2 in membrane trafficking in ASPS 17 cells (Fig. 5g, h, Supplementary Fig. 5g). Overall, these results indicate that intracytoplasmic trafficking of angiogenic factors, such as Pdgfb, Gpnmb, Angptl2, and Vwf, is promoted in ASPS cells and regulated by ASPSCR1::TFE3 via the Rab27a/Sytl2 axis.

**Fig. 3 | JQ1 suppresses angiogenesis-associated SEs and inhibits ASPS in vivo growth. a** Suppression of in vivo tumor growth by JQ1 treatment. Growth curves of the transplanted tumors treated with JQ1 or vehicle and the experimental schedule are shown (left) (*n* = 5 mice/10 independent tumors per group). The tumor volume was measured using 2 tumors per mouse. Gross appearances of tumors resected on day 17 (right). **b** Immunohistological examination of the tumor samples treated with JQ1 or vehicle. Anti-Cd31 (endothel), anti-Pdgfrb (pericyte) and anti-FLAG (tumor cell) were used. Cd31- and Pdgfrb-positive areas were measured and quantitated using the image J software. Positive areas were normalized by the number of FLAG-positive cells (right) (*n* = 17 and 5 independent areas for Cd31 and Pdgfrb, respectively). scale bar: 50 μm. Experiments represent three biological replicates. **c** GSEA showing PDGF-pathway enrichment by comparing vehicle-treated ASPS and ASPS treated with 0.5 μM JQ1 for 48 h gene expression signatures. The *p*-value is computed through the two-sided permutation test (*n* = 1000 randomizations) adjusted the Benjamini-Hochberg procedure. **d** Ingenuity pathway analysis (IPA) exhibiting multiple genetic pathways including angiogenesis/vasculogenesis (described in red) by JQ1 treatment. *p*-value was calculated using Fischer's exact test. **e** Composite plots showing a significant reduction in Brd4-binding signals around ASPSCR::TFE3-binding peaks by JQ1 treatment (left). Heat maps showing ASPSCR::TFE3 and Brd4 with or without JQ1 treatment, in murine ASPS17 cells (right). **f** Venn diagram showing overlapping and distinct Brd4-binding peaks in ASPS17 cells treated with JQ1 and vehicle (top). GREAT using 23,924 genetic loci specific for vehicle-treated ASPS17 showing the enrichment of angiogenesis-related pathways (bottom). *p*-values are calculated using a binominal test. **g** ChIP-seq track at *Rab27a, Sytl2, Pdgfb, Vwf, Myh9*, and *Neat1* genomic loci with ASPSCR1::TFE3-binding peaks in ASPS17 cells. Significant reduction of Brd4 signals at *Rab27a, Sytl2, Pdgfb*, and *Vwf* loci. ASPSCR1::TFE3 signals remained unchanged by JQ1 treatment. **h** qRT-PCR showing downregulation of *Rab27a, Sytl2, Pdgfb*, and *Vwf* in ASPS17 treated with JQ1, with the expression of *Myh9* and *Neat1* remaining unchanged (*n* = 3 per group). Statistical analyses in (**a, b, h**) were performed by two-sided Student's *t*-test and *marks adjusted *p*-value < 0.05, **marks adjusted *p*-value < 0.01 and ***marks adjusted *p*-value < 0.001. ns: no significance. The data presented as mean ± SD. Source data are provided as a Source Data file.

## *Rab27a* or *Sytl2* loss affects vascular sprouting in the microfluidic device

To confirm the significant role of the Rab27a/Sytl2 axis in angiogenesis, three-dimensional (3D) tissue culture/microfluidic devices were used to mimic in vivo angiogenesis. The microfluidic device included a tumor spheroid comprising murine ASPS cells covered with pericytes to constitute "core shell" structures (Fig. 6a, b, c, Supplementary Fig. 6a). Spheroid growth varied, despite the equal growth property and ASPSCR1::TFE3 expression of each ASPS clone (Supplementary Fig. 6a, b). The tumor spheroids were subsequently co-cultured with human umbilical vein endothelial cells (HUVEC) to evaluate the on-chip vascular extension. During 5 days of co-culturing, significant HUVEC sprouting toward the tumor spheroids of ASPS cells occurred; however, the sprouting effect was significantly suppressed by *Rab27a* or *Sytl2* knockout and *ASPSCR1::TFE3*-expression loss (Fig. 6d, e, Supplementary Fig. 6c). Decreases in vascular components of *Rab27a* or *Sytl2* knockout cells and ASPS null cells were also confirmed by immunohistochemical spheroid analysis (Fig. 6f). These results suggest that ASPSCR1::TFE3 induces endothelial sprouting via the Rab27a and Sytl2 axis in ASPS cells in collaboration with pericytes.

## Discussion

A prominent vascular network is a hallmark of ASPS, both in morphology and biological behavior. Abundant capillary blood vessels in ASPS are associated with its slow growth and resistance to conventional chemotherapy. The vascular structure of ASPS is reproducible in both genetically engineered mouse and xenograft models transplanted with human ASPS cells[17,18,22]. Our study demonstrated that angiogenic promotion occurs at the early stage of tumor growth in vivo (Fig. 1f, Supplementary Fig. 1e), indicating that this mechanism is essential for the ASPS development.

Previous studies identified numerous ASPSCR1::TFE3 targets that are involved in growth signaling, lysosomal functions, autophagy, and angiogenesis in both ASPS and fusion-positive renal cell carcinoma, with the latter exhibiting a vascular structure similar to that of ASPS[21,22,24,25,36]. Many of these genes are believed to be involved in the important biological processes of ASPS and tumorigenesis. However, the reported targets are rather diverse, probably due to differences in model systems and the lack of appropriate biological evaluation. Furthermore, in vitro studies using cell lines are considerably limited, and the clarification of angiogenic mechanisms requires both in vivo studies and advanced co-culture techniques, such as the organ-on-a-chip system used in this study. Significant tumor-growth inhibition in vivo by *ASPSCR1::TFE3* loss indicates the importance of co-proliferation between tumor cells and blood vessels in ASPS. The present study demonstrates the essential role of the ASPSCR1::TFE3 driven pathways in tumor development and angiogenesis. The similar tumor suppression and angiogenesis phenotypes and gene expression profiles between *ASPSCR1::TFE3* loss and the JQ1 treatment suggests the critical role of SEs modulated by ASPSCR1::TFE3, although the mechanism by which Brd4-inhibition targets certain SEs awaits clarification. A previous study indicated that key SEs define cell lineage, stemness, plasticity, and differentiation, which are remodeled by pioneer transcription factors such as SOX9, in hair follicle stem cells[37]. ASPSCR1::TFE3, as a pioneer factor, may also play a key role in SE modulation during ASPS development and angiogenesis.

RAB27 and SYTL/SLP proteins collaborate to facilitate cytoplasmic trafficking of vesicles, secretory granules, and target proteins, such as membrane-bound receptors[38,39]. These molecules are important in non-neoplastic pathological conditions, such as degranulation of neutrophils, cytotoxic T-cells, and platelets[38,39]. A RAB27A genetic mutation has been reported in the congenital disorder Griscelli syndrome type 2, in which the patients exhibit albinism and severe immune deficiencies[34,40]. Although *RAB27A* and *SYTL2* expression is upregulated in multiple cancer types[41], the biological significance of this upregulation is not well understood. We previously identified *Sytl1*, a close homolog of *Sytl2*, as a direct target gene of Meis1 in AML[42]. Sytl1, a Rab27b partner in hematopoietic cells, facilitates CXCR4 membrane trafficking and promotes occupancy of the bone marrow niche by AML, indicating the importance of the Rab27/Sytl axis in the interaction between tumor cells and the microenvironment. In this study, we identified Rab27a and Sytl2 as angiogenesis promoting factors that are downstream targets of ASPSCR1::TFE3-regulated SEs. Rab27a and Sytl2 do not directly affect vascular network construction; instead, they increase the secretion of angiogenic factors that are abundant in ASPS cells. Rab27a and Sytl2 upregulation indicates the importance of targeting this pathway for effective ASPS therapy using specific inhibitors, such as Nexinhib 20. Thus, improved inhibitors of this pathway are promising therapeutic tools.

Multiple factors that promote angiogenesis were found to be upregulated in ASPS[17,18,21,43]. Our present study revealed important angiogenic factors, such as PDGFB, GPNMB, and ANGPTL2, which are upregulated in ASPS, and they are included in the cargo transported from the cytoplasm to the membrane by RAB27A and SYTL2. However, we did not identify all the components sufficient for angiogenesis in ASPS. Further studies are required to clarify the complete angiogenic process of ASPS. Nevertheless, this study revealed that PDGFB is important for pericyte migration and proliferation[10,11,44], a key vascular-network component in ASPS. Accumulation of PDGFRB-positive cells were observed in an early phase of ASPS transplantation, suggesting that pericytes may be promptly migrated. Although PDGFRB is also expressed in other mesenchymal cells such as fibroblasts, the mature ASPS lesion exhibits the presence of abundant pericyte components, strongly

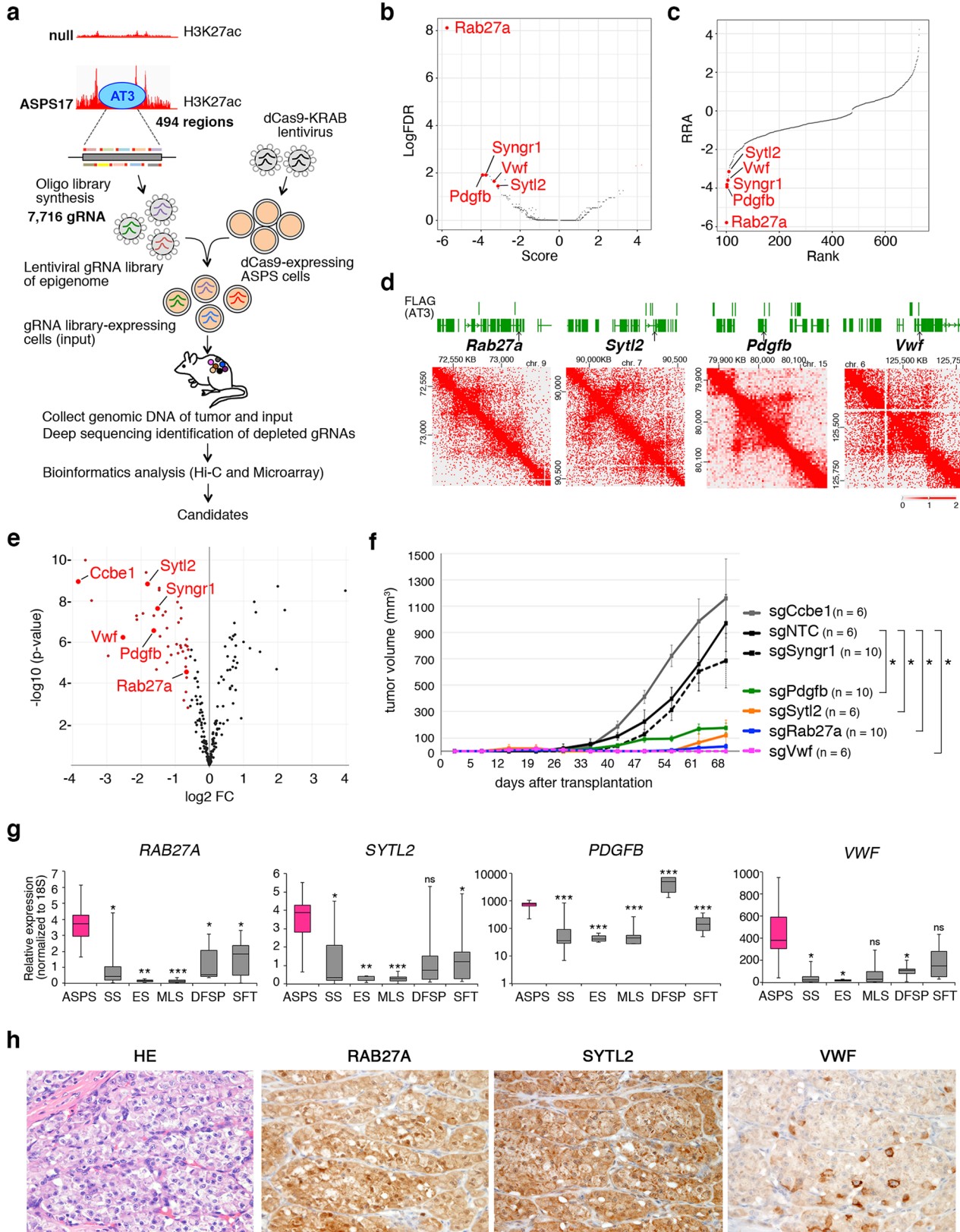

suggesting that certain fractions of PDGFRB-positive cells might contribute to ASPS vascular network. Moreover, GPNMB is essential for tumor-cell intravasation and extravasation in ASPS, given that *Gpnmb* knockdown significantly reduces the interaction between ASPS and endothelial cells[17]. Furthermore, GPNMB induces endothelial-cell migration[45], indicating its importance in tumor

vascular network formation. ANGPTL2 binds to endothelial cells and promotes vascular sprouting in collaboration with ANGPTL1[46,47].

VWF is a plasma glycoprotein that is predominantly produced by endothelial cells and megakaryocytes[48]. VWF is transported to Weibel–Palade bodies, and its secretion is promoted by Rab27a during the hemostatic process[49,50]. VWF also plays an important role in

**Fig. 4 | Identification of ASPSCR1::TFE3 targets by epigenomic CRISPR/dCas9 screening. a** Schematic illustration of the screening. ASPS17-specific super-enhancers and active enhancers positive for ASPSCR1::TFE3-binding peaks were selected for the construction of the gRNA library. ASPS17 cells were serially transduced with lenti-Ef1a-dCas9-KRAB-Puro and library lentivirus, and were transplanted into nude mice. Tumors were obtained 4 weeks after transplantation, and DNA samples were subjected to deep sequencing. Enhancers containing significantly decreased gRNA were identified and candidate target genes located within the same topologically associated domains (TADs) as enhancers were selected. After functional annotation, six genes were selected for in vivo validation. **b** Volcano plot illustrating the comparison of enhancer-associated genes whose repression is expected to inhibit in vivo tumor growth. **c** Illustration of candidate genes that are ranked based on the robust rank aggregation (RRA) score calculated by the MAGeCK algorithm. **d** Hi-C showing TADs at *Rab27a, Sytl2, Pdgfb*, and *Vwf* loci. The genetic lengths are labeled in each diagram. **e** Volcano plot comparing the expression of 271 candidate genes between ASPS17 and ASPS null cells. Dots labeled in red were determined as having a $\log_2$ (fold-change) of $\leq -1$ in null cells and an adjusted $p$-value $\leq 0.01$ (Bonferoni correction). **f** Growth curves of ASPS17 transplanted tumors indicating deleted genes ($n = 3$ mice/6 independent tumors per group for sgNTC, sgCcbe, sgSytl2 and sgVwf tumors, and 5 mice/10 independent tumors per group for sgSyngr1, sgPdgfb and sgRab27a tumors). The tumor volume was measured using 2 tumors per mouse. **g** qRT-PCR of *RAB27A, SYTL2, PDGFB*, and *VWF* in human sarcoma patients. ASPS: alveolar soft part sarcoma ($n = 7$), SS: synovial sarcoma ($n = 7$), ES: Ewing sarcoma ($n = 5$), MLS: myxoid liposarcoma ($n = 6$), DFSP: dermatofibrosarcoma protuberance ($n = 5$), SFT: solitary fibrous tumor ($n = 7$). The boxes show the interquartile range, with the median marked as a horizontal band. Whiskers represent the highest (lowest) datapoint within 1.5 times the interquartile range of the 75th (25th) percentile. The dots represent each datapoint. **h** Immunostaining of RAB27A, SYTL2, and VWF in human ASPS. Scale bar, 50 μm. Experiments represent three biological replicates. Statistical analyses in **f** were performed by two-sided Student's *t* test and in **g** were performed by one-way ANOVA. *marks adjusted $p$-value < 0.05, **marks adjusted $p$-value < 0.01 and ***marks adjusted $p$-value < 0.001. The data presented as mean ± SD. Box plots presented as median ± interquartile range.

---

angiogenesis and it functions as a reservoir of angiogenic growth factors[51]. Although the role of VWF secreted from ASPS cells remains unclear, angiogenic factors, such as PDGFB and VEGF, possibly bind to VWF to facilitate interactions between pericytes and endothelial cells. Indeed, a previous study demonstrated that VWF deficiency in mice causes a delay in angiogenesis[51].

Finally, we evaluated the importance of the ASPSCR1::TFE3 and Rab27a/Sytl2 axis using an angiogenesis model. As ASPS is characterized by an integrated vascular network comprising tumor cells, pericytes, and endothelial cells as described above, it is difficult to evaluate the modulation of angiogenic potential using a standard cell culture. The organ-on-a-chip system, using microfluidic devices and tumor spheroids, greatly improves monolayer-culture deficit[52]. In this study, we created a core shell structure containing ASPS tumor cells covered with pericytes, reflecting the alveolar structure of ASPS in vivo[19]. Proper and efficient interaction between ASPS and pericytes induces endothelial-cell sprouting. Significant ASPSCR1::TFE3, Rab27a, and Sytl2 contributions were highlighted in the induction of vascular sprouting. Inhibitory drugs for angiogenesis in ASPS would be evaluated using this system, enabling the development of improved and effective ASPS therapies. Rab27a-mediated transportation of cytoplasmic vesicles is involved in multiple cellular functions and strict inhibition of the function may induce unexpected effects, nevertheless, development of novel inhibitors will benefit the patients of refractory diseases such as ASPS.

## Methods

All research complies with relevant ethical regulations, including approval by Japanese Foundation for Cancer Research Institutional Review Board (IRB) under license 2013-1155 and the animal care committee at the Japanese Foundation for Cancer Research under licenses 10-05-9 and 0604-3-13.

### Cell lines and culture
The mouse ASPS cell lines ASPS17 and ASPS25 were established from the tumors induced in embryonic mesenchymal cells expressing *ASPSCR1::TFE3* as previously described[17]. The human ASPS cell lines, ASPS-KY and ASPS1, are described previously[53,54]. Mouse pericytes were purified from the dpc 17 mouse embryo mesenchyme by sorting the Pdgfrb+, Cd105+, and Cd31- fraction, and the cells were immortalized by introduction of the SV40 large T antigen. Aska and Yamato were obtained from Norifumi Naka[55]. KH was established in our lab[56]. A673, U2OS and HEK293T were purchased from ATCC (CRL-1598, HTB-96, and CRL-3216). ASPS-KY, ASPS1, Aska, Yamato, and KH were authenticated by the STR assay. None of cell lines used were under the list of known misidentified cell lines provided from the International Cell Line Authentication Committee (ICLAC). Mouse and human ASPS cell lines were grown in IMDM and RPMI1640 supplemented with 10% FBS, respectively. Mouse pericytes, Aska, Yamato, U2OS, and HEK293T cell lines were grown in DMEM with 10% FBS. A673 and KH were grown in RPMI1640 with 10% FBS. Absence of mycoplasma contamination was confirmed at regular intervals by PCR (Takara).

### Allograft transplantation studies
Tumor cell transplantation experiment was performed by injecting $1 \times 10^6$ ASPS cells mixed in Matrigel (Corning) into the subcutaneous regions of 6- to 8-weeks old female Balb/c nude mice. JQ1 and THZ1 were intraperitoneally administered daily for two weeks (50 mg of JQ1 and 10 mg of THZ1). Mice were carefully observed daily, and were euthanized 17 or 25 days after transplantation of ASPS17 or ASPS-KY cells, respectively. All animal experiments described in this study were performed in strict accordance with standard ethical guidelines under the 12 h lights on/12 h lights off cycle and at 23 °C and 50% humidity. Mice were euthanized by $CO_2$ inhalation before tumors reached a maximum size of 2000 mm³. The studies of ours and others demonstrated that no sex predisposition in the incidence and prognosis of alveolar soft part sarcoma, validating our initial choice to use only one sex. The experiments were approved by the animal care committee at the Japanese Foundation for Cancer Research under licenses 10-05-9 and 0604-3-13.

### Immunoblotting
Cells were lysed in RIPA buffer and precleaned by centrifugation at $10,000 \times g$ for 10 min at 4 °C. Protein concentrations were measured by the DC protein assay (Bio-Rad). Equal amout of protein lysates were boiled for 5 min in sample buffer (0.5 M dithiothreitol, 25% glycerol, 2% sodium dodecyl sulfate (SDS), 60 mM Tris-HCl pH 6.8 and bromophenol blue. The samples were separated by sodium dodecyl-sulfate polyacrylamide gel electrophoresis (SDS-PAGE) on SuperSep gels (Fuji Film) and transferred onto nitrocellulose membrane (Amersham). Immunoblots were probed with primary antibodies in 5% skim milk overnight at 4 °C and respective secondary antibodies for 1 h at room temperature. Primary antibodies used were ASPSCR1 (1:1000, Sigma-Aldrich, HPA026749), Cas9 (1:1000, Novus Biologicals, NBP2-36440), c-Myc (1:1000, Cell Signaling, 5605), CCBE1 (1:1000, Affinity Biosciences, DF10092), SYNGR1 (1:1000, Cell Signaling, 20874), PDGFB (1:1000, Abcam, ab23914), VWF (1:1000, Bioss, bs4754R), RAB27A (1:1000, Cell Signaling, 69295), SYTL2 (1:1000, Proteintech, 12359), FLAG-tag (1:1000, Sigma-Aldrich, F3165), Myc-tag (1:1000, Santa Cruz Biotechnology, sc-40), GFP (1:1000, Merck Millipore, MAB3580), mCherry (1:1000, Cell Signaling, 43590), DsRed (1:1000, Clontech, 632496), a-Tubulin (1:1000, Sigma-Aldrich, T5168), and Gapdh (1:1000, HyTest, 5G4). Rabbit IgG conjugated with HRP (1:2000, Cytiva, NA934, 1:2000) was used as a secondary antibody.

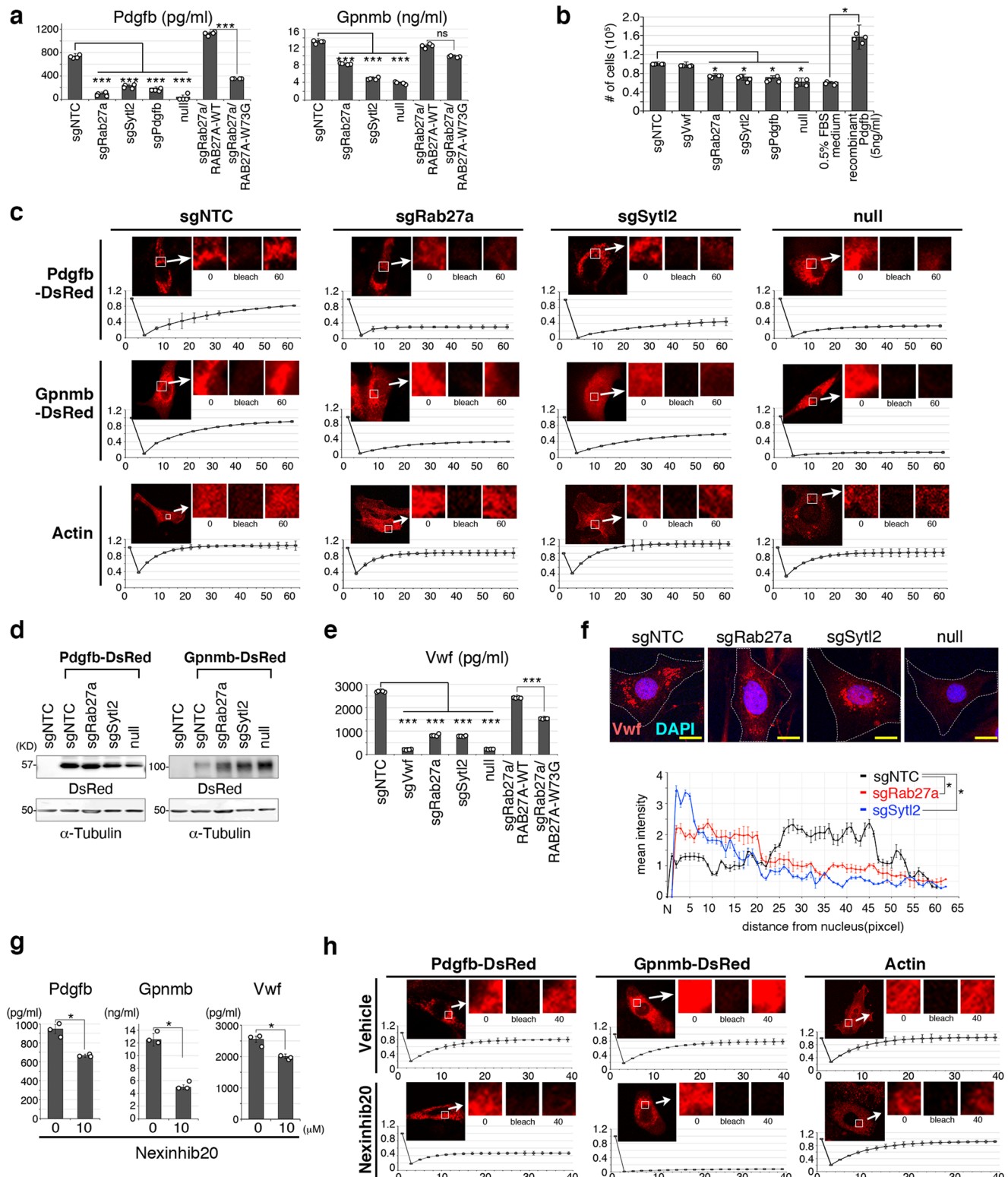

## Immunofluorescence

Mouse ASPS cells were fixed with 4% paraformaldehyde and were subjected to immunofluorescence using the specific antibodies and the respective fluorochrome-labeled secondary antibodies. Immunofluorescent images were photographed with a Zeiss LSM 710 laser scanning microscope with a 60x objective (Zeiss) and LSM Software ZEN 2009 (Zeiss). Captured images were analyzed using ImageJ. The mean distance of Vwf from the nucleus was calculated in 10 different cells for each condition per experiment by tracing lines (6 in each cell) from the nuclear membrane to the most distal fluorescent foci. For evaluation of 3D-culture in the microfluidic device, fluorescence and immunofluorescence images were captured using Olympus IX71 and Olympus FV3000, and recorded every other day and quantified as described in the figure legends. Antibodies used were VWF (not diluted, DAKO, N1505), FLAG-tag (1:100, Sigma-Aldrich, F7425), and goat anti-rabbit IgG conjugated with RRX (1:100, Jackson Immunoresearch, 111-295-144).

**Fig. 5 | Intracellular trafficking of angiogenic factors is promoted by the Rab27a/Sytl2 axis. a** Secretion of Pdgfb and Gpnmb in conditional media of ASPS17 cells measured by ELISA ($n = 4$ per group). Wild type *RAB27A* cDNA and the W73G mutant cDNAs were introduced. **b** Induction of pericyte growth promotion by an ASPS17-conditioned medium was inhibited by *Rab27a*, *Sytl2*, or *Pdgfb* knockout and *ASPSCR1::TFE3* loss. Cells were treated for 48 h ($n = 4$ per group). **c** Fluorescence recovery after photobleaching (FRAP) showing a marked delay in the trafficking of DsRed-labeled Pdgfb or Gpnmb by *Rab27a* knockout, *Sytl2* knockout, and ASPS null cells. Representative FRAP images are shown in the upper panels, and representative normalized traces of FRAP for each experiment are shown in the lower panels ($n = 4$ per group). Each point represents one image acquired every 5 min. RFP-tagged actin was used as a negative control. The data presented as mean ± SD. **d** Western blotting showing expression of DsRed-tagged Pdgfb and Gpnmb in each genotype. Data are presented by three independent experiments. **e** Vwf secretion in conditional media of ASPS17 cells measured by ELISA ($n = 4$ per group). **f** Immunofluorescence showing intracellular localization of Vwf. Localization in peripheral areas of ASPS17 cells treated with negative-control sgRNA was observed, whereas perinuclear Vwf localization was indicated by *Rab27a* or *Sytl2* knockout, and ASPS null cells (top). Intracellular Vwf distribution in each cell type was plotted as mean intensities of fluorescence at indicated distances from the nucleus (bottom) ($n = 4$ per group). Scale bar, 10 μm. Experiments represent three biological replicates. **g** Suppressed secretion of Pdgfb, Gpnmb and Vwf by Nexinhib 20 measured using ELISA ($n = 3$ per group). **h** FRAP showing significant delay in Pdgfb and Gpnmb trafficking by 10 μM of Nexinhib 20 treatment ($n = 3$ per group). The data presented as mean ± SD. Statistical analyses in (**a**, **b**, **e**) were performed by one-way ANNOVA and in (**f**, **g**) were performed by Student's *t* test and *marks adjusted *p*-value < 0.05, **marks adjusted *p*-value < 0.01, ***marks adjusted *p*-value < 0.001 and ns marks no significance. The data presented as mean ± SD.

## Quantitative RT-PCR (qRT-PCR)
Total RNA extraction, reverse transcription and RNA quantification were performed by standard methods. Conventional RT-PCR and real-time quantitative RT-PCR (Q-RT-PCR) were performed with a Gene Amp 9700 thermal cycler (Applied Biosystems) and a 7500 Fast Real-Time PCR System (Applied Biosystems), respectively. The sequences of the oligonucleotide primers are shown in Supplementary Data 12.

## Histopathology and immunohistochemistry
For light microscopic analysis, tumor tissues were fixed with 3% formaldehyde, paraffin embedded, and stained with hematoxylin and eosin (H&E) using standard techniques. Antibodies used were mouse CD31 (1:100, Cell Signaling, 77699), PDGFRB (1:100, R & D Systems, BAF1042), RAB27A (1:100, Cell Signaling, 69295), SYTL2 (1:100, Santa Cruz Biotechnology, sc393847), VWF (1:100, Santa Cruz Biotechnology, sc365712), human CD31 (1:100, Abcam, ab28346), FLAG-tag (1:100, Sigma-Aldrich, F3165), aSMA (1:100, DAKO, MO851), TFE3 (1:100, Santa Cruz Biotechnology, sc5958), and NG2 (1:100, Millipore, AB5320). Heat-mediated antigen retrieval was performed in Tris-EDTA buffer at pH 6.0. Immunohistochemical staining was performed using the Simple Stain MAX-PO kit (Nichirei Bioscience), the Histofine SAB-PO (R) kit (Nichirei Bioscience).

## Gene expression profiling
ASPS cell pellets were processed to extract total RNeasy Mini Kit (Qiagen). RNA quality was assessed using the Bioanalyzer 2100 (Agilent). The murine Genome HT MG-430 PM Array and the human HG-U133+ PM Array (Affymetrix, Santa Clara, CA, USA) were hybridized with aRNA probes generated from the total RNA samples. After staining with streptavidin-phycoerythrin conjugates, arrays were scanned using an Affymetrix GeneAtlas Scanner.

## Microarray data processing and analysis
The data were analyzed with Microarray Suite version 5.0 (MAS 5.0) using Affymetrix default analysis settings and global scaling as normalization method. The trimmed mean target intensity of each array was arbitrarily set to 100. Microarray data were analyzed using GeneSpring GX 14.9 (Agilent). Gene pathway analysis was performed using Gene Set Enrichment Analysis (GSEA) was performed using GSEA-P 2.0 software[57], and Ingenuity Pathway Analysis (Qiagen).

## RNA sequencing
Total RNA was prepared (Qiagen) and quality checked. The RNA-seq libraries were made using SMARTer Stranded Total RNA-seq Kit (Takara Bio). Next generation sequencing was performed on an Illumina NextSeq 500.

## RNA-seq analysis
To generate expression count matrix, row reads were trimmed to remove adaptor sequences by Skewer (v0.2.2). UMIs were extracted by umi_tools 'extract' and UMI adaptor sequences were also removed by cutadapt (v1.15). The processed reads were mapped to hg38 genome by STAR (v.2.7.8a), and then de-duplicated by umi_tools 'dedup'. The mapped reads were counted by featureCounts (v.2.0.10). edgeR's glmQLFTest (v3.32.1) was used to identify differential gene expression. We used as input groups of control and intervention experiments with a simple design with a 0 intercept '-0 + Group'. First, library sizes were normalized by calculating scaling factors with 'calcNormFactors (y, method = TMM)'. Dispersions were then estimated with 'estimateDisp (y, design = design, robust = TRUE)', and fitted the generalized linear model using 'glmQLFit (y, design = design)'. Finally, log2 fold-changes and false-discovery rates were calculated by glmQLFTest.

## Chromatin immunoprecipitation (ChIP)-sequencing
ChIP-Seq was performed using the method previous described with modifications[58]. A total of $5 \times 10^6$ ASPS cells per immunoprecipitation were cross-linked with 1% formaldehyde for 10 min at room temperature. Chromatin was sheared in SDS lysis buffer containing 1% SDS, 10 mM EDTA, and 50 mM Tris pH 8.0 to an average size of 400 to 500 bp using a Covaris S220 sonicator for 15 min. ChIP was performed with 5 μg anti-histone H3K27ac (Active Motif, 39133), anti-H3K4me3 (Abcam, ab8580), anti-H3K27me3 (Millipore, 07-449), anti-FLAG (Sigma-Aldrich, F7425), anti-ASPSCR1 (Sigma-Aldrich, A026749), anti-BRD4 (Bethyl Laboratories, A301-985A100) antibodies. The antibody-bound protein/DNA complexes were immunoprecipitated using protein G magnetic beads. Immunoprecipitated DNA was then purified and subjected to secondary sonication to an average size 150 to 350 bp. Libraries were prepared according to instructions accompanying the ThruPLEX DNA-Seq kit (Rubicon Genomics). The ChIP DNA was end modified and adapters were ligated. DNA was PCR amplified with Illumina primers and Illumina-compatible indexes were added. The library fragments of approximately 300–500 bp were band-isolated from an agarose gel. The purified DNA was sequenced on an Illumina MiSeq next-generation sequencer following the manufacturer protocols.

## ChIP-seq data analysis
Base calls were performed using Bowtie 2 (http://bowtie-bio.sourceforge.net/bowtie2/index.shtml). ChIP-Seq reads were aligned to the mm9 (https://www.ncbi.nlm.nih.gov/assembly/GCF_000001635.18) or hg19 (https://www.ncbi.nlm.nih.gov/assembly/GCF_000001405.13/) genome assembly using samtools 1.2 (http://www.htslib.org). Peak calling was performed using MACS1.4 (http://liulab.dfci.harvard.edu/MACS). Peak distribution was calculated by Cistrome (http://cistrome.org/ap/root). Neighbor genes on enriched genomic regions were determined using by Nucleus (https://rias.rhelixa.com). The genomic distributions of DNA-binding peaks were visualized by NGSplot (https://anaconda.org/bioconda/r-ngsplot). DNA-binding of each ChIP-seq data was visualized using IGV_2.3.80 (http://software.broadinstitute.org/software/igv). The de novo motif

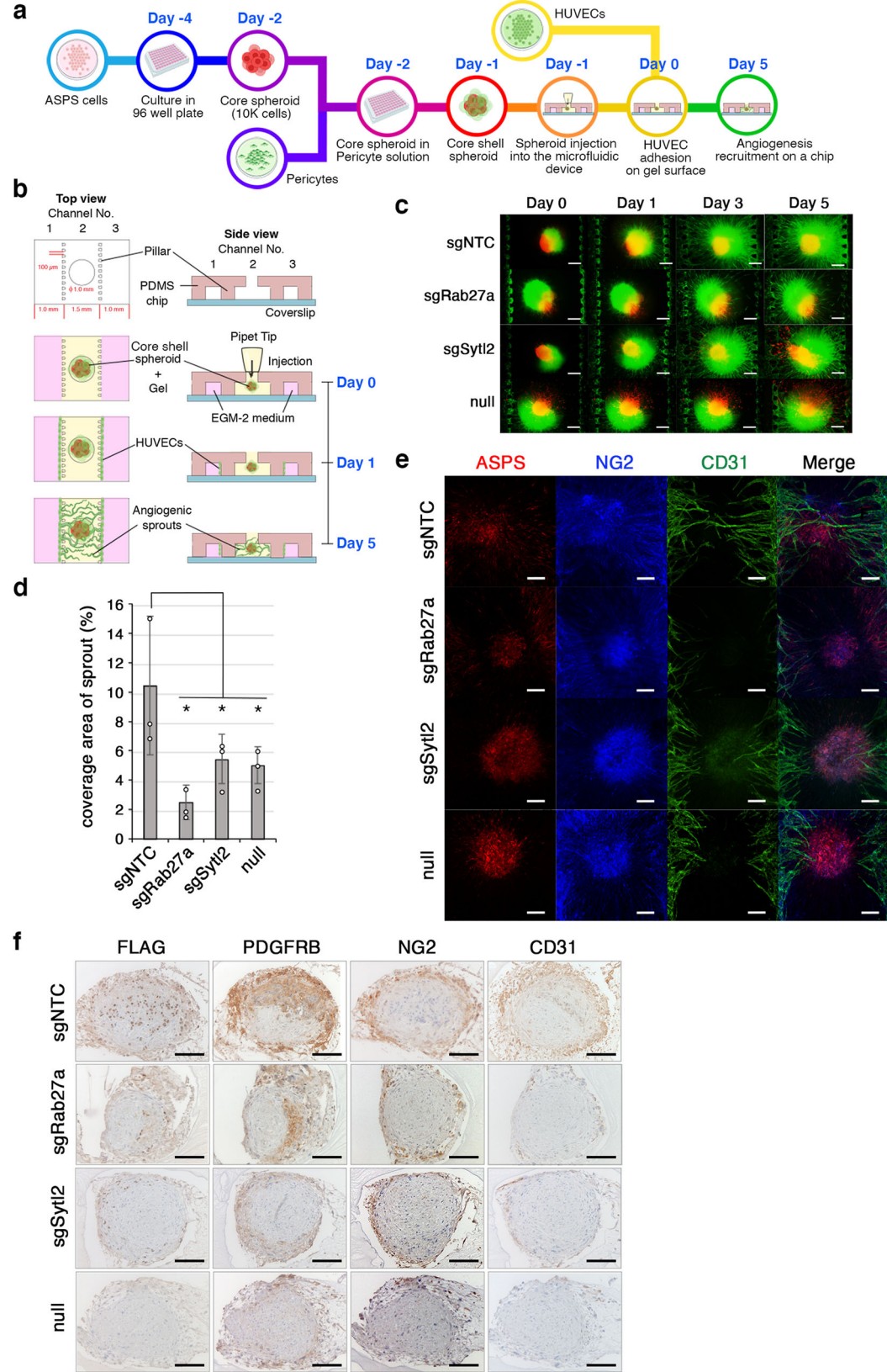

enrichment was performed using HOMER v 4.11.1 (http://homer.ucsd.edu/homer/motif). Super-enhancers were identified using the method previously described with the ROSE program (http://younglab.wi.mit.edu/super_enhancer_code.html)[27,59]. Gene ontology analysis for nearby genes on super-enhancers were performed by GREAT version 4.0.4 (http://great.stanford.edu/public/html/).

**Epigenomic CRISPR/dCas9 screening**

CRISPR/Cas9-based epigenomic screening was performed according to the method described with modifications[60,61]. Total of 494 SEs and active enhancers in which H3K27ac signals were reduced more than 40% by *ASPSCR1::TFE3* loss were selected, and average 12.8 gRNAs per each enhancer were designed to target these enhancers using CRISPR

**Fig. 6 | Induction of vascular network by co-culture of tumor spheroids, pericytes, and HUVECs in the microfluidic device. a** Schematic representation of the in vitro vascular network experiment. **b** The structure of spheroid formation on the three-channel microfluidic device. **c** Core shell spheroids composed of ASPS17 cells (red) and human pericytes (green), and HUVECs (green) in the microfluidic device. The time-course fluorescent images of the in vitro angiogenesis using the core-shell ASPS spheroids (sgNTC, sgRab27a, sgSytl2, and null) on days 0, 1, 3, and 5. Experiments represent three biological replicates. **d** Quantification for surface coverage of angiogenic sprouts on day 5 (*n* = 3 per group). The statistical analysis was performed by one-way ANOVA, *p* < 0.05. The data are represented as the mean ± SD. **e** Representative images at day 5 clearly show the presence of angiogenic sprouts by CD31 staining and pericytes by NG2 staining. Scale bar, 200 μm. Experiments represent three biological replicates. **f** Immunohistochemical analysis of spheroids showing the expression of ASPSCR1::TFE3 (FLAG) for ASPS cells, PDGFRB and NG2 for pericytes, and CD31, for HUVECs. Increase in PDGFRB- and NG2-positive pericytes and CD31-positive HUVECs in sgNTC spheroids. Scale bar, 100 μm. Experiments represent three biological replicates.

direct (https://crispr.dbcls.jp) and GuideScan (http://www.guidescan.com) (Supplementary Data 8). 397 gRNAs for enhancers without ASPSCR1::TFE3 binding and 1000 non-target gRNAs (https://www.addgene.org/pooled-library/zhang-mouse-gecko-v2/) were designed as negative controls. Gibson overhangs were fused to sense and anti-sense nucleotides corresponding to each gRNA. The gRNA library was synthesized by CustomeArray and inserted into pLV-U6-gRNA-UbC-DsRed-P2A-Bsr (Addgene) using Gibson Assembly (New England Bio-Labs). A high titer gRNA library was constructed in Endura Electro-Competent cells (Lucigen), and 293FT cells were transfected with the purified library DNA, psPAX2 and pCMV-VSV-G. ASPS17/dCas9-KRAB cells were generated by infecting Lenti-Ef1a-dCas9-KRAB-Puro lentivirus (Addgene) and the cells were transduced with lentivirus bearing the gRNA library at 100x fold coverage. DsRed-positive cells were sorted by FacsAria II and selected using 2 μg/mL of Blasticidin four days after infection. Then, $1 \times 10^6$ infected cells were transplanted sub-cutaneously to nude mice 10 days after infection. Subcutaneous tumors were removed a month after transplantation and genomic DNA was extracted, PCR amplified, and subjected to target sequencing using screening primers listed in Supplementary Data 13.

### Data analysis of CRISPR/dCas9 screening
gRNAs were mapped to the annotation file (0 mismatch), and read count tables were made. Sequencing reads were compared between pre-transplanted and post-transplanted ASPS 17 cells using DESeq2. MAGeCK[31] was used to normalize the count tabled based on median normalization and fold changes, and the significance of differences was calculated. Downstream statistical analyses and generating plots were performed using MAGeCKFlute[62].

### Hi-C
The Hi-C libraries were constructed using an Arima-HiC Kit (Arima Genomics) according to the manufacturer's instructions for Mammalian Cell Lines (A160134 v00) and Library Preparation (A160137 v00). In brief, one million cells were collected and crosslinked with 37% for-maldehyde solution. DNA isolated from the crosslinked cells was digested with two restriction enzymes (^GATC and G^ANTC). After the incorporation of biotinylated nucleotides at the digested DNA ends, both ends were ligated with the spatially proximal ends. The ligated DNA was sheared into 200–600 bp fragments using the Focused-ultrasonicator M220 (Covaris) and the ligation junctions were enriched with streptavidin magnetic beads. The sequencing libraries from enriched DNA fragments were prepared with a TruSeq DNA PCR-Free Library Prep Kit (Illumina). The resulting libraries were amplified with 10 PCR cycles and purified with SPRI beads. The quality and con-centration of the paired-end libraries were evaluated using the Qubit 4 Fluorometer (Thermo Fisher Scientific), the 2100 Bioanalyzer system (*Agilent* Technologies), and the 7900HT Fast Real-Time PCR System (Thermo Fisher Scientific). The final libraries were sequenced on the Illumina NovaSeq 6000 sequencer with a read length of 150 bp. Reads were processed with Juicer pipeline, using reference of hg38 for human samples and mm10 for mouse samples, yielding.*hic* files[63]. These files were further used for Juicebox visualization and down-stream analysis, including finding loops and contact domains using Juicer tools. Loops were called by HICCUPS and those located in Chr1-22 (human), 1–19 (mouse), and X.

### CRISPR/Cas9-mediated gene editing
Lentivirus plasmids containing short guided RNA (sgRNA) for *Ccbe1*, *Pdgfb*, *Rab27a*, *Syngr1*, *Sytl2*, and *Vwf* were introduced into the lentiCRISPRv2-puro vector (Addgene). The sgRNA sequences are listed in Supplementary Data 13. Knockdown efficiencies were confirmed by western blotting and/or RT-PCR.

### Enzyme-linked immunosorbent assay (ELISA)
For the ELISA, $5 \times 10^4$ mouse ASPS cells were seeded and cultured overnight. The culture media were collected and immediately analyzed using the mouse Gpnmb (Abcam), Vwf (Abcam), and Pdgfb (Pro-teintech) ELISA kits according to manufacturers' instructions.

### Fluorescence recovery after photobleaching
Trafficking of Pdgfb, Gpnmb, and Angptl2 was evaluated by FRAP following previously described procedures[64]. ASPS17 and null cells were transfected with with plasmids containing, DsRed-tagged Pdgfb, Gpnmb, or Angptl2. RFP-tagged actin (Invitrogen) was used as a negative control of trafficking. FRAP images were acquired with an LSM880 confocal microscope equipped with a live cell chamber (set at 37 °C) and ZEN software (Zeiss) with a 40X objective. Cells were exci-ted with a 561 nm laser and the emission between 566 and 689 nm recorded. Images were acquired with 12 bits image depth and 512 × 512 resolution using a pixel dwell of ~1.52 μs. At least three ($n \geq 3$) pre-bleaching images were collected and then the region of interest was bleached with 100% of laser power. The recovery of fluorescence was traced at every 5 min for 1 h. For drug inhibition experiments, 10 μM of Nexinhib 20 (Tocris) was added into culture media 3 h before bleaching. Fluorescence recovery rates were corrected for internal photobleaching and background, and they were normalized to pre-photobleaching intensities as previously described[62].

### Fabrication of microfluidic device
The micropatterned master with 100 μm in height features was fabricated based on standard photolithography. SU-8 3050 (Micro-Chem) was coated and patterned on a 4 inch silicon wafer by exposing ultraviolet (UV) light to the top of a photomask on which the desired 3-channel patterns are printed. A mixture of PDMS (Sylgard 184, Dow-Corning) and its curing agent was mixed to a 10:1 weight ratio, poured onto the patterned master to a 5-mm thickness, degassed to remove air bubbles for 1–2 h, and cured in an oven at 80 °C for 3–4 h. The cured PDMS was removed from the wafer. The inlets (diameter: 2 mm), outlets (diameter: 6 mm), and center hole (diameter: 1 mm) for media filling and spheroid injection were punched using biopsy punches unless otherwise noted. After cleaning with adhesive tape, the PDMS device and glass coverslip were permanently bonded using oxygen plasma treatment (40 s, 50 sccm, 40 mW). Furthermore, the microfluidic devices were then stored in a 35 mm dish and cured in an oven at 80 °C for more than 2 h. After sterilization under UV for 1–2 h, the microfluidic devices were ready for cell seeding.

## ASPS spheroid formation

The prewarmed DMEM was added to the dish, and the ASPS cell suspensions were transferred into 15 mL tubes and centrifuged at 1000 rpm for 3 min. The cells were resuspended at desired density. Cell suspension was then added in a prime surface 96-well plate with U-shaped bottom well (Sumitomo Bakelite), which significantly caused self-aggregation of cells, namely a spheroid. Monoculture core spheroids were prepared with a density of $5.0 \times 10^4$ cells/mL in the IMDM medium for 2 days. Co-culture core-shell spheroids were initiated by core spheroid formation. After two days, the suspension culture of core spheroids was then replaced by human placental microvascular pericytes (Angio-Proteomie) suspension at a density of $7.5 \times 10^4$ cells/mL in the IMDM medium for shell formation for another day. Finally, the spheroids were introduced into a micro-fluidic device.

## In vitro angiogenesis

A spheroid was transferred into fibrin-collagen gel (2.5 mg/mL fibrinogen) (Sigma-Aldrich), 0.15 U/mL aprotinin (Sigma-Aldrich), 0.2 mg/mL collagen type I (Corning), and thrombin 0.5 U/mL (Sigma-Aldrich) in D-PBS. The spheroid suspended in gel solution was then injected through the center hole into channel 2 without leakage into channels 1 and 3 and incubate at a 37 °C $CO_2$ incubator for 15 min for gelation. Channels 1 and 3 were filled with EGM-2 and incubated overnight at the incubator to eliminate the bubbles at the boundary of gel and medium. For HUVEC adhesion at the gel interface, the HUVECs ($5.0 \times 10^6$ cells/mL in the EGM-2) were injected into channel 1. With a 90° tilt device at a 37 °C $CO_2$ incubator for 15 min, HUVECs adhered to the gel surface in the microfluidic device. This process was repeated for channel 3. After spheroid injection and HUVEC adhesion, the inlets and outlets of channels 1 and 3 were filled with EGM-2 and kept at 37 °C and 5% $CO_2$ in an incubator. For culturing cells on the device, EGM-2 was replaced daily unless otherwise noted.

## Human sarcoma specimens

Alveolar soft part sarcoma, synovial sarcoma, Ewing sarcoma, myxoid liposarcoma, dermatofibrosarcoma protuberans and solitary fibrous tumor surgical specimens were obtained from The Cancer Institute Hospital. All samples were collected from 37 patients (20 females and 17 males, 5 to 87 years of age). Informed written consent was obtained from all the patients, and the study was approved by Institutional Review Board at the Japanese Foundation for Cancer Research under license 2013-1155. Patients were consecutively included in this study, therefore there were no selection of patients based on sex in this study. Twenty females and 17 males were included in this study. Gender of participants was not intentionally selected, since did not find association of the sex and gene expression profiles in sarcoma patient samples.

## Data comparisons in human sarcoma samples

Six microarray studies containing 166 tumor samples that were analyzed using U133 Plus 2.0A Array (Affimetrix) were queried for gene expression. CEL files from GSE12102, GSE13433, GSE20196, GSE32569, GSE66533, and E-MTAB-1361 were downloaded. The probe sets for *RAB27A, SYTL2, PDGFB*, and *VWF* were examined.

## Statistics and data reproducibility

Statistical analyses were performed using Microsoft Excel for Mac Version 16. A two-tailed Student *t*-test was used when comparing two groups and one-way ANOVA statistical method was used for multiple comparison. The mean ± SD of individual experiments is shown. All immunofluorescence, immuhohistochemistry, qRT-PCR and western blotting results were repeated at least twice using independently prepared samples, and similar results were obtained.

## Reporting summary

Further information on research design is available in the Nature Portfolio Reporting Summary linked to this article.

## Data availability

The raw RNA-seq and ChIP-seq data generated in this study have been deposited in the NCBI Gene Expression Omnibus (GEO) database (http://www.ncbi.nlm.nih.gov/geo) under the accession number GSE215265 and GSE189163, respectively. The raw Hi-C data generated in this study have been deposited in the DDBJ Sequenced Read Archive under the accession numbers DRR330423 and DRR330424. The reused publicly available data for gene expression are available in the NCBI Gene Expression Omnibus (GEO) database (http://www.ncbi.nlm.nih.gov/geo) under the accession number GSE12102[65], GSE13433[21], GSE20196[66], GSE32569[14], GSE66533[67], and in the EMBL-EBI ArrayExpress (https://www.ebi.ac.uk/biostudies/arrayexpress) under the accession number E-MTAB-1361[68]. The following human genome accessions were used: The NCBI accession of the UCSC hg19 genome is GCA_000001405.1. The following mouse genome accessions were used: GRCM37/mm9 assembly via UCSC under GCA_000001635.1. The remaining data are available within the Article, Supplementary Information or Source Data file. Source data are provided with this paper.

## Materials availability

Plasmids generated in this study are available upon request (T.N.)

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

## Acknowledgements

The authors thank Rikuka Shimizu, Yasuyo Teramura, Tomoko Takahara, Liyin Yang, and Hinako Ishizaki for technical assistance, Ryohei Nakamura for data evaluation, and Robert Shoemaker for providing ASPS1 cells. This study was supported by Grants-in-Aid for Scientific Research from the Japan Society for the Promotion of Science (16K07131 and 19K07702 to M.T., and 26250029 to T.N.), and by a Grant-in-Aid Project for Cancer Research and Therapeutic Evolution from the Japan Agency for Medical Research and Development (17cmA106609 to T.N., 20cm0106277 to T.N. and R.Y., 21ak0101170 to T.N., and 22ama221206 to M.T. and R.Y.). This work was also supported by JSPS KAKENHI Grant Number 16H06279 (PAGS).

## Author contributions

M.T.: investigation, formal analysis, data curation, visualization, validation, funding acquisition and writing-original draft. S.C.: data curation, investigation, methodology and visualization. M.H.: investigation and validation. Y.Y.: investigation. R.L.: investigation. K.Y.: resources. K.A.: resources. S.M.: resources. K.K.: data curation and formal analysis. R.M.: data curation and formal analysis. W.Q.: formal analysis and data curation. Y.M.: resources. R.Y.: methodology, conceptualization, funding acquisition and supervision. T.N.: conceptualization, investigation, project administration, supervision, funding acquisition, writing—original draft and editing.

## Competing interests

The authors declare no competing interests.
