## [Peer Review File · Nature Communications]

ASPSCR1-TFE3 orchestrates the angiogenic program of alveolar soft part sarcomaReviewers' Comments:

Reviewer #1:

Remarks to the Author:

In this work Tanaka et al used a wide array of experimental approaches to investigate the functional role of the ASPSCR1-TF3 (AT3) fusion protein in defining the biological properties of ASPS tumors. They combined expression and chromatin profiling with CRISPR screening and in vitro/in vivo functional assays to identify a new role for this fusion protein in shaping the vascular network that characterizes these aggressive malignancies. This group has a well-established track record in translocation-driven sarcomas in general, and ASPS in particular, providing them a solid knowledge and know-how in this field. The manuscript is well written, the experiments appropriately designed, and the conclusion robustly supported by the results. Altogether, this work provides new and interesting findings in the biology of ASPS tumors and beyond, since the angiogenic switch presented here may be relevant in other biological settings, and lead to new improved therapeutic strategies.

However, before considering this work for publication in Nature Communications, several points need to be addressed.

Major Points:

- It is unclear how many replicates of mouse and human ASPS tumors were used to generate the expression data presented in Figure 1. In order to improve the robustness and reproducibility of these results at least 2 different mouse models (AT3-expressing vs null) and 2 human ASPS lines (control vs AT3 KD) should be used. It would also be very valuable to present a Venn diagram describing the overlap of DE genes between all the models used, to provide a better understanding of the shared programs controlled by AT3.
- Fig1F: it is surprising that this analysis was performed at such an early time point (4 days after transplantation). At this time point it may be difficult to fully assess the impact of the tumor cells on the microenvironment, including angiogenesis. Could the authors replicate the analyses showed in Fig 1F at a later time point?
- The chromatin profiling shown in Fig 2 is key for all subsequent analyses of this study. As such, this profiling should be reinforced by adding several replicates, and assessing the reproducibility of the AT3 peak call in several models. To this end the authors should perform at least 2 replicates for the Flag and ASPSCR1 ChIP-seq, and present data about their overlap, as well as provide a clear statement of how the AT3 peaks were defined in Supplementary Figure 2.
- Along the same line, the authors should validate the specificity of their ChIP-seq signal and AT3 peak set by repeating the Flag and ASPSCR1 profiles in mouse (AT3-expressing Vs AT3 null) and human (hASPS control vs AT3 KD) tumor cells. This is a key point: if the AT3 signal they have identified and used to define their peak set is specific, the vast majority should strongly decrease in AT3 null or KD cells.
- Fig 2D: it is unclear how the different heatmaps were generated. How was the signal ranked? In addition, looking at the composite plots for H3K27ac signal in hASPS it seems that the decrease in KD cells is minimal, whereas the corresponding heatmaps seem to show a greater difference. This should be reevaluated.
- The authors should provide a genome-wide analysis of K27ac levels (heatmaps and composite plots) showing that the decrease in H3K27ac signal they observe at the AT3 binding sites is not a reflection of a more general decrease in this histone mark due to cells suffering from the loss of AT3. To this end the authors should check the H3K27ac levels on all TSSs genome-wide, and compare AT3-proficient to depleted cells (mouse and human).

- Fig 3: the authors provide interesting data about the effect of JQ1 on ASPS tumor cells, however the current results are focused on the mouse model, limiting their impact for human tumors. To expand these findings the authors should also generate the same data using at least one human ASPS line.
- Fig 3E: this is a very interesting finding that once again should be also confirmed using human tumor cells. In addition, it will be key to assess in both mouse and human in vivo models if the administration of JQ1 also impacts the AT3 binding profile. To this end the authors should repeat the AT3 ChIP-seq on in vivo treated and control mouse and human ASPS tumors, and check if they observe a change in the AT3 binding profile that could explain the changes in BRD4 signal and gene expression that follow JQ1.
- Figure 4F: these results are very compelling. Is there a more extensive gene expression database of sarcomas, that also includes ASPS, and that the authors could use to extend their analysis to a higher number of tumor samples?
- Fig 6E: the KO of Rab27a and Syt12 also seems to strongly reduce the Flag IHC signal. The authors should check by qPCR and Western Blot if the KO of these two genes also reduces the expression levels of AT3.

Minor Points:

- The authors should describe the expression of the wild type ASPSCR1 and TFE3 proteins in their mouse and human tumor models. When using an ASPSCR1 antibody to define the AT3 binding profile it is key to make sure they are not detecting wild type ASPSCR1.
- Lines 210-211: the authors should revise their sentence that currently states that 7716 gRNAs were used for each selected enhancer.
- Line 216: there is a typo.
- Lines 251 to 253: the authors to rephrase this sentence, which is currently unclear.

Reviewer #2:

Remarks to the Author:

In my opinion, this manuscript is mainly based on a previously established ectopic expression model of retroviral ASPSCR1-TFE3 fusion cassette to generate ASPS tumors. They revealed the inactivation of fusion proteins in vitro for an unknown reason and further identified its requirement for in vivo tumor maintenance. They claimed that ASPSCR1-TFE3 fusion protein is essential for maintaining super-enhancer activity in selected genes responsible for many pathways, including angiogenesis. Blocking BRD4 and enhancer activity by JQ1 or knocking out the angiogenesis-related target genes can significantly retard tumor progression in vivo. To my knowledge, ASPSCR1-TFE3 fusion protein has been identified as driver mutations in ASPS and other solid tumors, including renal carcinoma, which predicts poor prognosis in patients. There is an urgent need to understand the molecular mechanism underlying ASPSCR1-TFE3 biology. The study falls into two parts as interrogating enhancer regulation control of ASPSCR1-TFE3 by conducting AF3, H3K27ac ChIP-seq, RNA-seq, and JQ1 treatment in mouse and human ASPS cancer cell lines and attempting to reveal the role of angiogenesis in the ASPSCR1-TFE3-driven in vivo models. However, the main conclusion was not

sufficiently supported data presented. In particular, the mouse model using retroviral and ectopic expression brings a significant concern to me (will explain in detail below). Enclosed, please find my significant concerns regarding scientific rigor and data interpretation that may hamper publishing this manuscript at the current status. I hope my comments below will help the authors strengthen the finding further.

1. I have significant concerns about the mouse tumor models. The author noticed the silencing of AT3 during long-term in vitro culture, which is probably due to the viral cassette silencing related to methylation, mutation, or random insert sites. I agree the tumorigenesis model of the initial Cancer Research paper in 2017 provides a proof-in-principle for modeling ASPS. However, to study AT3 biology, a more human-relevant model is better. For instance, human patient-derived xenograft or CRISPR-mediated in vivo mouse translocation AT3 model will maintain the endogenous regulatory mechanism intact rather than constitutively retroviral LTR. Alternatively, mouse tumor models with AT3 tagged with the dTAG or degron system are minimally required (see below). I think these models should validate the main conclusion.

2. RNA-seq and ChIP-seq characterization in this study more likely define the endpoint difference of two cell states rather than the causal changes. This is due to the nature of crude knockout by siRNA. It is essential to know the primary function of AT3 and connect the regulation with enhancer activity maintenance and angiogenesis regulation. To this end, the acute protein depletion model using dTAG and degron will serve as the best tools. The main conclusion regarding gene expression change and enhancer activity change should be repeated using the acute depletion model to confirm the finding further.

3. In figure 3, the heatmap of AT3 ChIP-seq peaks displays a similar pattern in selected loci shown in Figure 2G, suggesting AT3 fusion protein likely occupied the DNA as broader peaks. However, general TF binding patterns, including bHLH TFs, should represent sharp peaks. Based on ours and many other reports, motif enrichment is not ideal for broad peak ChIP-seq data. Given that I do not have access to evaluate the data quality, I will strongly suggest the authors provide more QC data of these ChIP-seq in figures or supplementary materials. What is the average size of the AT3 peak? What's the motif enrichment% in for the targeted sequences vs. background? In Figure 2D, the demonstration of the heatmap is confusing. It is difficult to compare the K27ac activity with AT3 peak affinity. The authors should interpret the heatmap in a better way. Again, with the dTAG or degron AT3 model, the authors will define the true peaks using isogenic knockout systems.

4. In Figure 3, whether the JQ1 effect is through inactivating MYC expression rather than erasing genome-wide enhancer activity? It can be explained by a rescue experiment with ectopic expression of MYC in a JQ1 treated setting.

5. In Figure 4, there is a limited description of the genome-wide analysis of CRISPR-dCas9 screens of the noncoding H3K27ac marked regions. The authors should provide more data as MeGACK analysis and ranking data.

Reviewer #3:

Remarks to the Author:

This is an extremely interesting manuscript that describes the orchestration of the angiogenic program of alveolar soft part sarcoma by epigenetically regulated ASPSCR1-TFE3.

This reviewer is not an expert on epigenetics or some of the chip-seq types of technology used in this study and will focus on the angiogenesis aspects of the work

Overall is clear that loss of ASPS17 can induce a significant antiangiogenic phenotype.

Figure 1

Apart from the loss of CD31 there is a significant loss of PDGFRB expression. It is not appropriate to assume that PDGFRB expression is pericyte restricted here since it may well be expressed on other stromal cells. This is of particular interest in that it suggests a desmoplastic response regulated by ASPSCR. Have the authors considered this further? Minimally some expanded discussion on this would make the work of interest to a wider audience. There is clearly a 'chicken and egg' effect with loss of blood vessel having an effect on desmoplasia, but arguably the opposite scenario also may be relevant and it would be interesting to have some discussion on this. Since all PDGFRB is not restricted to blood vessels and may be found in fibroblasts their involvement in the angiogenic regulation should be discussed.

Is it correct that the histology, described in F, is really only 4 days after cell injection? Given that there is no tumour apparent at day 7 post injection in the null or ASP17 groups is the data describing an inflammatory response to the injection rather than the basis of tumour angiogenesis. Perhaps histology from 11 days would be better?

Fig 6

The reviewer commends the authors for the use of the microfluiding technology to examine angiogenesis. The authors state that spheroid growth varied, despite the equal growth property of each ASPS clone (Supplementary Fig. 6A). Does this not imply that the stimulus of angiogenesis is varied? This reviewer was therefore not confident that the changes in angiogenesis were not due to changes in tumour cell proliferation and respective secretion of proangiogenic stimuli. Can the authors explain or provide data to indicate that the changes in spheroid growth are not reflected in the levels of angiogenesis induced? This reviewer also does not understand why the null cells do not appear to express FLAG-tag in D or E? What was the rationale for labelling both the pericytes and HUVECs in GFP?? This method means that it is impossible to ascertain the involvement of the proportions of HUVECs and pericytes in the vessel formation. Staining for PDGFRB in 'E' does not look like perivascular cells. Are the authors sure that they are not EMT of the tumour cells? In 'A' there is no indication of the addition of HUVECs to the system and this would be good to include please.

Translation

The authors touch briefly on how their finding may have translational value. Although I strongly disagree that all discovery science should have a translational angle it would be nice to see some further elaboration on the pros and cons of the discovered mechanisms in a translation discussion.

Reviewer #4:

Remarks to the Author:

Tanaka, et al. have produced an exciting interrogation of one angiogenesis pathway targeted for upregulation by ASPSCR1-TFE3 through superenhancers in alveolar soft part sarcoma, a rare, but deadly malignancy with young victims. Overall, the investigation is rich in the depth to which it probes four gene targets of the fusion oncoprotein. There are even clinically translatable insights gained from some of the testing with JQ1. A few major weaknesses and minor weakness could be readily addressed in a revision.

MAJOR WEAKNESSES

1. The overall premise of the work that ASPSCR1-TFE3 is dispensable to proliferation, but not tumor growth is founded on essentially no evidence for the first of those statements. The authors note that there was "frequent" loss of ASPSCR1-TFE3 expression during in vitro passage of their mouse

engineered cell lines, but only demonstrate one example. Some data regarding a larger sample size or even noting the denominator of how many times they observed it in their other studies would be helpful. Also, the very weak knock-down achieved in ASPS-KY cells, that already grow at painfully slow rates, might demonstrate a "false-negative" in terms of the effect on proliferation. (What are the units of the "cell growth rate") in Fig1d? Certainly, ASPS-KY has not seen two doublings in 3 days for 4X+ fold-change; they hardly double in a week.)

2. At the end of all of this there is very shallow rigor in the genomics and other experiments, almost all of which were performed with a single murine cell line and a second "control" murine cell line with the lost expression of ASPSCR1-TFE3. What is the sample size for each expression experiment or ChIP-seq experiment. I cannot find these sample sizes anywhere. The use of singular siRNAs for each target in each cell line is also concerning. Usually, at least 2 would be necessary to prove that it is not some off-target effect. This is especially important when the knock-down of ASPSCR1-TFE3 is so slight at about 75% message loss and only a little more than 50% loss of protein. These same principles apply to the other target gene knock-downs.

3. More clarity is necessary in how the analysis was pared down to these few angiogenic genes. Were there comparative genomics performed between ASPS-KY siNTC/siTFE3 and ASPS17/Null? The SuppFig4D comes the closest to explaining it, but even that is not entirely clear. We just need to see how you got to these particular genes, as many other angiogenesis genes are also likely impacted to render those DAVID analysis results.

MINOR WEAKNESSES

1. The nomenclature for mouse proteins throughout is incorrectly using non-italicized gene symbols with only the first letter capitalized. While this in italics is the correct nomenclature for mouse genes, the corresponding gene products should be noted with non-italicized ALL CAPS symbols, as conforming to accepted guidelines for mouse proteins.

2. There are few typos scattered throughout, some of which are important, such as labeling THZ1 as "JQ1" in the supplemental Figure 3E legend.

3. Please don't switch to mASPS/hASPS labeling from ASPS17/ASPS-KY labeling in Fig1B+D. Just note the cell line names to keep it all clear.

4. In general, using RT-PCR alone as confirmation of knock-down of a given gene, rather than showing western blot seems somewhat less than optimal (Supplemental Fig4E)

5. While the FRAP experiments are insightful with regard to movement of PDGFB and GPNMB through the cells, you should confirm that these knock-downs don't also change the expression levels of these proteins.

RESPONSE TO THE REVIEWER

We are grateful to the reviewer for the critical comments and useful suggestions that have helped us to improve our manuscript. As indicated in our responses below, we have taken all of your comments and suggestions into account in this revised version of our paper.

Authors' responses to reviewers are indicated in blue.

Reviewer #1, expertise in sarcoma, functional genomics, Hi-C, ChIP-seq (Remarks to the Author):

In this work Tanaka et al used a wide array of experimental approaches to investigate the functional role of the ASPSCR1-TF3 (AT3) fusion protein in defining the biological properties of ASPS tumors. They combined expression and chromatin profiling with CRISPR screening and in vitro/in vivo functional assays to identify a new role for this fusion protein in shaping the vascular network that characterizes these aggressive malignancies. This group has a well-established track record in translocation-driven sarcomas in general, and ASPS in particular, providing them a solid knowledge and know-how in this field. The manuscript is well written, the experiments appropriately designed, and the conclusion robustly supported by the results. Altogether, this work provides new and interesting findings in the biology of ASPS tumors and beyond, since the angiogenic switch presented here may be relevant in other biological settings, and lead to new improved therapeutic strategies.

However, before considering this work for publication in Nature Communications, several points need to be addressed.

Response: Thank you, we greatly appreciate your positive response.

Major Points:

- It is unclear how many replicates of mouse and human ASPS tumors were used to generate the expression data presented in Figure 1. In order to improve the robustness and reproducibility of these results at least 2 different mouse models (AT3-expressing vs null) and 2 human ASPS lines (control vs AT3 KD) should be used. It would also be very valuable to present a Venn diagram describing the overlap of DE genes between all the models used, to provide a better understanding of the shared programs controlled by AT3.

Response: We have added mouse and human ASPS cell lines, ASPS25 and ASPS1, and gene expression profiles and growth properties have been investigated. The data are presented in Fig. 1b and Supplementary Fig. 1c. A Venn diagram describing the overlap of differentially

expressed genes and significant association with characteristic pathways such as exosome transport and melanosome have been shown in Supplementary Fig. 1i. Moreover, similar AT3 knockdown effects on gene expression in the ASPS1 cell is shown in Supplementary Fig. 1h.

- Fig1F: it is surprising that this analysis was performed at such an early time point (4 days after transplantation). At this time point it may be difficult to fully assess the impact of the tumor cells on the microenvironment, including angiogenesis. Could the authors replicate the analyses showed in Fig 1F at a later time point?

Response: Induction of blood vessels and the tumor microenvironment by ASPS is rapid, and there is no significant differences of angiogenesis between day 4 and 14. The data have been added in Supplementary Fig. 1d.

- The chromatin profiling shown in Fig 2 is key for all subsequent analyses of this study. As such, this profiling should be reinforced by adding several replicates, and assessing the reproducibility of the AT3 peak call in several models. To this end the authors should perform at least 2 replicates for the Flag and ASPSCR1 ChIP-seq, and present data about their overlap, as well as provide a clear statement of how the AT3 peaks were defined in Supplementary Figure 2.

Response: We have repeated ChIP-seq experiments for AT3 peaks in both mouse and human ASPS cells. Peak distributions, motif analysis, and GO annotation have been performed using peaks common in two independent ChIP-seq experiments. These data are presented in Fig. 2a-d and Supplementary Fig. 2a-c.

- Along the same line, the authors should validate the specificity of their ChIP-seq signal and AT3 peak set by repeating the Flag and ASPSCR1 profiles in mouse (AT3-expressing Vs AT3 null) and human (hASPS control vs AT3 KD) tumor cells. This is a key point: if the AT3 signal they have identified and used to define their peak set is specific, the vast majority should strongly decrease in AT3 null or KD cells.

Response: We appreciate the reviewer's advice. We performed ChIP-seq for FLAG and ASPSCR1 in mouse and human ASPS cells, respectively, and confirmed that the peak signals are specific and most of peaks disappeared or reduced. There remain some peaks in siTFE3-treated human ASPS-KY cells probably due to incomplete knockdown effects, however, the reduction of signals is obvious. These data have been added in Fig. 2d, g and Supplementary Fig. 2g.

- Fig 2D: it is unclear how the different heatmaps were generated. How was the signal ranked?

In addition, looking at the composite plots for H3K27ac signal in hASPS it seems that the decrease in KD cells is minimal, whereas the corresponding heatmaps seem to show a greater difference. This should be reevaluated.

Response: We have corrected the previous version of heatmaps in which ranking of genomic regions was measured by hierarchical clustering. In the new version regions are ranked by the sum of the enrichment values, and the difference of H3K27Ac signals corresponds to that in the composite plot. The data are shown in Fig. 2d and Supplementary Fig. 2c.

- The authors should provide a genome-wide analysis of K27ac levels (heatmaps and composite plots) showing that the decrease in H3K27ac signal they observe at the AT3 binding sites is not a reflection of a more general decrease in this histone mark due to cells suffering from the loss of AT3. To this end the authors should check the H3K27ac levels on all TSSs genome-wide, and compare AT3-proficient to depleted cells (mouse and human).

Response: As described above, we have corrected heatmaps of K27ac peaks in Fig. 2d and Supplementary Fig. 2c. In addition, total numbers of K27ac peaks are compared between AS17 and null cells in Supplementary Fig. 2b, indicating that there is no significant reduction of total H3K27ac signals.

- Fig 3: the authors provide interesting data about the effect of JQ1 on ASPS tumor cells, however the current results are focused on the mouse model, limiting their impact for human tumors. To expand these findings the authors should also generate the same data using at least one human ASPS line.

Response: According to the reviewer's suggestion, we treated human ASPS-bearing nude mice with JQ1, and found that the growth suppression and inhibition of angiogenesis was similar as those in mouse ASPS. In addition, BRD4 binding at ASPSCR1::TFE3 target gene loci was decreased by JQ1. These results have been added in Supplementary Fig. 3a, b, d, g, and h.

- Fig 3E: this is a very interesting finding that once again should be also confirmed using human tumor cells. In addition, it will be key to assess in both mouse and human in vivo models if the administration of JQ1 also impact the AT3 binding profile. To this end the authors should repeat the AT3 ChIP-seq on in vivo treated and control mouse and human ASPS tumors, and check if they observe a change in the AT3 binding profile that could explain the changes in BRD4 signal and gene expression that follow JQ1.

Response: According to the reviewer's recommendation, JQ1 treatment was performed for human ASPS cells. We observed similar growth suppression and angiogenesis inhibition as shown in Supplementary Fig. 3a, 3b. It was difficult to perform ChIP-seq analysis in vivo tumor

samples. Instead, we have added the ChIP-seq results for human ASPS cells and AT3 binding profiles of both human and mouse cells are exhibited. These data are now shown in Fig. 3g, Supplementary Fig. 3d, g, and h.

- Figure 4F: these results are very compelling. Is there a more extensive gene expression database of sarcomas, that also includes ASPS, and that the authors could use to extend their analysis to a higher number of tumor samples?

Response: Thank you for the valuable suggestion. We obtained gene expression data from the GEO database, and the upregulated expression of RAB27A, SYTL2, PDGFB, and VWF in human ASPS among 5 different sarcoma types. The data is shown in Supplementary Fig. 4i.

- Fig 6E: the KO of Rab27a and Sytl2 also seems to strongly reduce the Flag IHC signal. The authors should check by qPCR and Western Blot if the KO of these two genes also reduces the expression levels of AT3.

Response: Expression of ASPSCR1::TFE3 at both RNA and protein levels in spheroids of all genotypes has been added in Supplementary Fig. 6b.

Minor Points:

- The authors should describe the expression of the wild type ASPSCR1 and TFE3 proteins in their mouse and human tumor models. When using an ASPSCR1 antibody to define the AT3 binding profile it is key to make sure they are not detecting wild type ASPSCR1.

Response: ASPSCR1 is a ubiquitously expressed gene encoding a UBX domain-containing protein that interacts with glucose transporters. ASPSCR1 is a cytoplasmic protein that does not contain putative DNA-binding motifs (Bogan et al, Nature, 2003; Cloutier et al, PLoS Genet, 2013; Habtemichael et al, Nat Metab, 2021). Therefore, we believe that the anti-ASPSCR1 antibody does not detect wild type DNA-binding signals in our ChIP-seq experiments.

- Lines 210-211: the authors should revise their sentence that currently states that 7716 gRNAs were used for each selected enhancer.

Response: We apologize the misinterpretation. The sentence has been corrected (lines 220-221).

- Line 216: there is a typo.

Response: This sentence has been replaced with new one.

• Lines 251 to 253: the authors to rephrase this sentence, which is currently unclear.

Response: This sentence has been edited appropriately (lines 266 to 267).

Reviewer #2, expertise in ChIP-seq, CRISPR (Remarks to the Author):

In my opinion, this manuscript is mainly based on a previously established ectopic expression model of retroviral ASPSCR1-TFE3 fusion cassette to generate ASPS tumors. They revealed the inactivation of fusion proteins in vitro for an unknown reason and further identified its requirement for in vivo tumor maintenance. They claimed that ASPSCR1-TFE3 fusion protein is essential for maintaining super-enhancer activity in selected genes responsible for many pathways, including angiogenesis. Blocking BRD4 and enhancer activity by JQ1 or knocking out the angiogenesis-related target genes can significantly retarded tumor progression in vivo. To my knowledge, ASPSCR1-TFE3 fusion protein has been identified as driver mutations in ASPS and other solid tumors, including renal carcinoma, which predicts poor prognosis in patients. There is an urgent need to understand the molecular mechanism underlying ASPSCR1-TFE3 biology. The study falls into two parts as interrogating enhancer regulation control of ASPSCR1-TFE3 by conducting AF3, H3K27ac ChIP-seq, RNA-seq, and JQ1 treatment in mouse and human ASPS cancer cell lines and attempting to reveal the role of angiogenesis in the ASPSCR1-TFE3-driven in vivo models. However, the main conclusion was not sufficiently supported data presented. In particular, the mouse model using retroviral and ectopic expression brings a significant concern to me (will explain in detail below). Enclosed, please find my significant concerns regarding scientific rigor and data interpretation that may hamper publishing this manuscript at the current status. I hope my comments below will help the authors strengthen the finding further.

Response: We appreciate your valuable comments.

1. I have significant concerns about the mouse tumor models. The author noticed the silencing of AT3 during long-term in vitro culture, which is probably due to the viral cassette silencing related to methylation, mutation, or random insert sites. I agree the tumorigenesis model of the initial Cancer Research paper in 2017 provides a proof-in-principle for modeling ASPS. However, to study AT3 biology, a more human-relevant model is better. For instance, human patient-derived xenograft or CRISPR-mediated in vivo mouse translocation AT3 model will maintain the endogenous regulatory mechanism intact rather than constitutively retroviral LTR. Alternatively, mouse tumor models with AT3 tagged with the dTAG or degron system are minimally required (see below). I think these models should validate the main conclusion.

Response: We understand the reviewer's points that PDX models, CRISPR-mediated translocation model or the use of the dTAG system might provide better information. However, we believe that the present study in this paper using our ex vivo model that faithfully recapitulates human ASPS phenotypes succeeded to identify key angiogenesis molecules in ASPS and novel insight into ASPS biology.

2. RNA-seq and ChIP-seq characterization in this study more likely define the endpoint difference of two cell states rather than the causal changes. This is due to the nature of crude knockout by siRNA. It is essential to know the primary function of AT3 and connect the regulation with enhancer activity maintenance and angiogenesis regulation. To this end, the acute protein depletion model using dTAG and degron will serve as the best tools. The main conclusion regarding gene expression change and enhancer activity change should be repeated using the acute depletion model to confirm the finding further.

Response: We agree that rapid depletion of AT3 will may provide some information in chromatin remodeling. However, we do not consider that our results define the endpoint difference, because JQ1 treatment of both mouse and human ASPS resulted in similar growth suppression *in vivo* and enhancer modification at angiogenesis-related genetic loci. In addition, we repeated the experiments using additional mouse and human ASPS cell lines, and obtained similar results. These data have been demonstrated in Fig. 1, 2, Supplementary Fig. 1 and 2.

3. In figure 3, the heatmap of AT3 ChIP-seq peaks displays a similar pattern in selected loci shown in Figure 2G, suggesting AT3 fusion protein likely occupied the DNA as broader peaks. However, general TF binding patterns, including bHLH TFs, should represent sharp peaks. Based on ours and many other reports, motif enrichment is not ideal for broad peak ChIP-seq data. Given that I do not have access to evaluate the data quality, I will strongly suggest the authors provide more QC data of these ChIP-seq in figures or supplementary materials. What is the average size of the AT3 peak? What's the motif enrichment% in for the targeted sequences vs. background? In Figure 2D, the demonstration of the heatmap is confusing. It is difficult to compare the K27ac activity with AT3 peak affinity. The authors should interpret the heatmap in a better way. Again, with the dTAG or degron AT3 model, the authors will define the true peaks using isogenic knockout systems.

Response: Thank you for your advice. The average size of the AT3 peak, motif enrichment frequencies both in peaks and non-peaks have been described in supplementary materials. Although median peak width was 722 ~ 890 bp, the frequency of the TFE3 motif within peaks was significantly enriched (35.57% within peaks and 4.35% outside peaks). With these results we believe that our ChIP-seq data are reasonable. The data are described in the legend of Fig.

2b.

4. In Figure 3, whether the JQ1 effect is through inactivating MYC expression rather than erasing genome-wide enhancer activity? It can be explained by a rescue experiment with ectopic expression of MYC in a JQ1 treated setting.

Response: Thank you for the suggestion. We have examined MYC expression upon JQ1 treatment in human and mouse ASPS cells, and have confirmed that MYC remained unchanged by JQ1 treatment. The data is present in Supplementary Fig. 3j.

5. In Figure 4, there is a limited description of the genome-wide analysis of CRISPR-dCas9 screens of the noncoding H3K27ac marked regions. The authors should provide more data as MeGACK analysis and ranking data.

Response: According to the reviewer's comment, we re-analyzed our CRISPR/dCas9 screening using the MAGECK program and ranked enriched regions. Fig. 4b, d, Supplementary Fig. 4c, 4d, Supplementary Table 9 and 10 have been deleted or modified. The results are visualized using the MAGECKFlute software. We have inserted the data analysis of CRISPR screening section in Methods (lines 522-527), and the results are shown in Fig. 4b, 4c, 4e, Supplementary Fig. 4c, Supplemental Table 9, 10 and 11.

Reviewer #3, expertise in tumour angiogenesis (Remarks to the Author):

This is an extremely interesting manuscript that describes the orchestration of the angiogenic program of alveolar soft part sarcoma by epigenetically regulated ASPSCR1-TFE3.

This reviewer is not an expert on epigenetics or some of the chip-seq types of technology used in this study and will focus on the angiogenesis aspects of the work

Overall is clear that loss of ASPS17 can induce a significant antiangiogenic phenotype.

Response: Thank you, we greatly appreciate your positive response.

Figure 1

Apart from the loss of CD31 there is a significant loss or PDGFRB expression. It is not appropriate to assume that PDGFRB expression is pericyte restricted here since it may well be expressed on other stromal cells. This is of particular interest in that it suggests a desmoplastic response regulated by ASPSCR. Have the authors considered this further? Minimally some expanded discussion on this would make the work of interest to a wider audience. There is clearly a 'chicken and egg' effect with loss of blood vessel having an effect on desmoplasia, but arguably the opposite scenario also may be relevant and it would be interesting to have some

discussion on this. Since all PDGFRB is not restricted to blood vessels and may be found in fibroblasts their involvement in the angiogenic regulation should be discussed.

Response: Thank you for your valuable comments. We have added the possibility of other stromal cell reaction in discussion (lines 356-360).

Is it correct that the histology, described in F, is really only 4 days after cell injection? Given that there is no tumour apparent at day 7 post injection in the null or ASPS17 groups is the data describing an inflammatory response to the injection rather than the basis of tumour angiogenesis. Perhaps histology from 11 days would be better?

Response: Induction of blood vessels and the tumor microenvironment by ASPS is rapid, and there is no significant differences of angiogenesis between day 4 and 14. The data have been added in Supplementary Fig. 1d.

Fig 6

The reviewer commends the authors for the use of the microfluiding technology to examine angiogenesis. The authors state that spheroid growth varied, despite the equal growth property of each ASPS clone (Supplementary Fig. 6A). Does this not imply that the stimulus of angiogenesis is varied? This reviewer was therefore not confident that the changes in angiogenesis were not due to changes in tumour cell proliferation and respective secretion of proangiogenic stimuli. Can the authors explain or provide data to indicate that the changes in spheroid growth are not reflected in the levels of angiogenesis induced? This reviewer also does not understand why the null cells do not appear to express FLAG-tag in D or E? what was the rationale for labelling both the pericytes and HUVECs in GFP?? This method means that it is impossible to ascertain the involvement of the proportions of HUVECs and pericytes in the vessel formation. Staining for PDGFRB in 'E' does not look like perivascular cells. Are the authors sure that they are not EMT of the tumour cells? In 'A' there is no indication of the addition of HUVECs to the system and this would be good to include please.

Response: We appreciate your constructive comments. We do not have direct data on the relationship between tumor cell proliferation and proangiogenic stimuli, however, core spheroid growth is comparable in each genotype as shown in Fig. 6c and Supplementary Fig. 6a. There was incorrect interpretation for core and core-shell in the previous version of Supplementary Fig. 6a, and the labeling has been corrected. We apologize for it.

In null cells, expression of FLAG-tagged AT3 was lost as shown in Fig. 1, and it should not be expressed spheroids shown in Fig. 6f.

We stained pericytes with an anti-NG2 antibody in new Fig. 6e to distinguish pericytes from CD31-positive HUVECs clearly.

For pericyte detection, we have performed NG2 immunostaining in spheroid and have confirmed PDGFRB-positive cells are NG2-positive pericytes, not tumor cells with EMT. The result has been added in Fig. 6f.

HUVEC introduction has been illustrated in the new version of Fig. 6a.

Translation

The authors touch briefly on how their finding may have translational value. Although I strongly disagree that all discovery science should have a translational angle it would be nice to see some further elaboration on the pros and cons of the discovered mechanisms in a translation discussion.

Response: Thank you for your comment. Although we have already described a statement on development of novel anti-angiogenic therapies utilizing our present study in discussion (lines 347-348 and 382-383), one sentence has been added to mention the pros and cons in a translation (lines 383-386).

Reviewer #4, expertise in sarcoma, functional genomics including CRISPR and mouse models (Remarks to the Author):

Tanaka, et al. have produced an exciting interrogation of one angiogenesis pathway targeted for upregulation by ASPSCR1-TFE3 through superenhancers in alveolar soft part sarcoma, a rare, but deadly malignancy with young victims. Overall, the investigation is rich in the depth to which it probes four gene targets of the fusion oncoprotein. There are even clinically translatable insights gained from some of the testing with JQ1. A few major weaknesses and minor weakness could be readily addressed in a revision.

Response: Thank you, we greatly appreciate your positive response.

MAJOR WEAKNESSES

1. The overall premise of the work that ASPSCR1-TFE3 is dispensible to proliferation, but not tumor growth is founded on essentially no evidence for the first of those statements. The authors note that there was "frequent" loss of ASPSCR1-TFE3 expression during in vitro passage of their mouse engineered cell lines, but only demonstrate one example. Some data regarding a larger sample size or even noting the denominator of how many times they observed it in their other studies would be helpful. Also, the very weak knock-down achieved in ASPS-KY cells, that already grow at painfully slow rates, might demonstrate a "false-negative" in terms of the

effect on proliferation. (What are the units of the "cell growth rate") in Fig1d? Certainly, ASPS-KY has not seen two doublings in 3 days for 4X+ fold-change; they hardly double in a week.)

Response: We have shown the number of cell lines in which ASPSCR1::TFE3 expression disappeared in the text (line 119). We have corrected the y-axis description of cell growth curves to the number of cells in Fig. 1d and Supplementary Fig. 1d. In addition, we have repeated knockdown of ASPSCR1::TFE3 in human ASPS cells, and have shown more convincing results in Fig. 1d and Supplementary Fig. 1c.

2. At the end of all of this there is very shallow rigor in the genomics and other experiments, almost all of which were performed with a single murine cell line and a second "control" murine cell line with the lost expression of ASPSCR1-TFE3. What is the sample size for each expression experiment or ChIP-seq experiment. I cannot find these sample sizes anywhere. The use of singular siRNAs for each target in each cell line is also concerning. Usually, at least 2 would be necessary to prove that it is not some off-target effect. This is especially important when the knock-down of ASPSCR1-TFE3 is so slight at about 75% message loss and only a little more than 50% loss of protein. These same principles apply to the other target gene knock-downs.

Response: We have repeated experiments using two murine and two human ASPS cells. We have two replicates of ChIP-seq experiments, which has been described in the legend of Fig. 2a and Reporting Summary. We also used two independent siRNAs for knockdown as shown above.

3. More clarity is necessary in how the analysis was pared down to these few angiogenic genes. Were there comparative genomics performed between ASPS-KY siNTC/siTFE3 and ASPS17/Null? The SuppFig4D comes the closest to explaining it, but even that is not entirely clear. We just need to see how you got to these particular genes, as many other angiogenesis genes are also likely impacted to render those DAVID analysis results.

Response: We have changed the gene selection process in CRISPR/dCas9 screening by applying MAGeCK for gRNA evaluation and evaluation of upregulated genes in human ASPS samples obtained from GEO database (Supplementary Fig. 4c). As a result, The number of genes decreased to 25 and we confirmed 6 candidates were included after functional annotation.

MINOR WEAKNESSES

1. The nomenclature for mouse proteins throughout is incorrectly using non-italicized gene symbols with only the first letter capitalized. While this in italics is the correct nomenclature for

mouse genes, the corresponding gene products should be noted with non-italicized ALL CAPS symbols, as conforming to accepted guidelines for mouse proteins.

Response: We have corrected all protein and gene nomenclatures carefully.

2. There are few typos scattered throughout, some of which are important, such as labeling THZ1 as "JQ1" in the supplemental Figure 3E legend.

Response: We have corrected typos including "JQ1" in the legend of Supplementary Fig. 3I.

3. Please don't switch to mASPS/hASPS labeling from ASP17/ASP17-KY labeling in Fig1B+D. Just note the cell line names to keep it all clear.

Response: We have unified the cell name as ASP17 and ASP17-KY, and mASPS/hASPS have been omitted. In figures, ASP17 and ASP17 Null are indicated as ASP17 and null for clarity.

4. In general, using RT-PCR alone as confirmation of knock-down of a given gene, rather than showing western blot seems somewhat less than optimal (Supplemental Fig4E)

Response: Western blotting was performed to confirm the gene knockout. The results is shown in Supplementary Fig. 4e.

5. While the FRAP experiments are insightful with regard to movement of PDGFB and GPNMB through the cells, you should confirm that these knock-downs don't also change the expression levels of these proteins.

Response: We appreciate the reviewer's comment. Expression of DsRed-tagged Pdgfb and Gpnmb is shown in Fig. 5d.

Reviewers' Comments:

Reviewer #1:

Remarks to the Author:

I acknowledge the efforts made by the authors to answer all my points. They have provided convincing additional data confirming their previous finding.

Therefore, I do not have any additional question.

Reviewer #2:

Remarks to the Author:

MY comments have been reasonable addressed. Thanks.

Reviewer #3:

Remarks to the Author:

The authors have presented an interesting piece of work and all my comments from the first review have been satisfied.

Reviewer #4:

Remarks to the Author:

Thank you for your careful response to many of the items elevated to your attention on the primary reviews. I reiterate that the overall body of this work is interesting and exciting in its ultimate scope.

The primary weakness of the study is the initial premise. It seemed that every reviewer commented on this. There is nothing wrong with using cells that have naturally--over extended culture--silenced expression of the fusion oncogene as a control of sorts for the fusion expressing cells, but it is not tantamount to a knock-down of the fusion in those cells. It is not a very good control at all, if you are trying to focus on the effects of loss of AT3. Why don't you just use your siRNAs in one of the mouse lines. They should work, as you are expressing human AT3. Because the knock-down experiments in ASPS-KY and ASPS1 cell lines were not very robust, it makes all of that basic premise that loss of AT3 doesn't impact proliferation very difficult to believe. Maybe this whole section should merely be left out of the paper. It is not critical to the later dive into the angiogenesis target genes identified on the screen.

Other problems result from some of the in vivo work on vasculature (Fig 1f, SuppFig1d, Fig3b). While it is reported in your response to reviewers that there is no difference between 4 days and 14 days, the fact of the matter is that the vasculature patterns in none of those photomicrographs mimics ASPS vasculature (which you show nicely in Supp3b, actually). It also does not look like AT3 expression is the same on the FLAG-IHC photomicrographs in Fig3b.

Finally, the ChIP-seq in 2d fits with the suggestions in Fig1 and SuppFig1 that the knock-down was very poor of AT3. Really, a stronger knock-down should be achieved before trying to do ChIP-seq.

My recommendation would be to expand the explanation of the CRISPR-KRAB screen and begin with that, as opposed to all the efforts to prove that AT3 loss, which is very poorly controlled in your experiments, has no impact on proliferation and only on angiogenesis.

RESPONSE TO THE REVIEWER

We are grateful to the reviewer for the critical comments and useful suggestions that have helped us to improve our manuscript. As indicated in our responses below, we have taken all of your comments and suggestions into account in this revised version of our paper.

Authors' responses to reviewers are indicated in blue.

Reviewer #4, expertise in sarcoma, functional genomics including CRISPR and mouse models (Remarks to the Author):

Thank you for your careful response to many of the items elevated to your attention on the primary reviews. I reiterate that the overall body of this work is interesting and exciting in its ultimate scope.

Response: Thank you again, we greatly appreciate your positive response.

The primary weakness of the study is the initial premise. It seemed that every reviewer commented on this. There is nothing wrong with using cells that have naturally--over extended culture--silenced expression of the fusion oncogene as a control of sorts for the fusion expressing cells, but it is not tantamount to a knock-down of the fusion in those cells. It is not a very good control at all, if you are trying to focus on the effects of loss of AT3. Why don't you just use your siRNAs in one of the mouse lines. They should work, as you are expressing human AT3. Because the knock-down experiments in ASPS-KY and ASPS1 cell lines were not very robust, it makes all of that basic premise that loss of AT3 doesn't impact proliferation very difficult to believe. Maybe this whole section should merely be left out of the paper. It is not critical to the later dive into the angiogenesis target genes identified on the screen.

Response: Thank you for your advice. We have performed siRNA-mediated knockdown of ASPSCR1::TFE3 in murine ASPS17 and ASPS25 cells, and confirmed that the decreased expression did not affect cellular growth significantly. The results are described in the text (lines 123-125) and Supplementary Figure 1d.

Other problems result from some of the in vivo work on vasculature (Fig 1f, SuppFig1d, Fig3b). While it is reported in your response to reviewers that there is no difference between 4 days and 14 days, the fact of the matter is that the vasculature patterns in none of those photomicrographs mimics ASPS vasculature (which you show nicely in Supp3b, actually). It also does not look like AT3 expression is the same on the FLAG-IHC photomicrographs in Fig3b.

Response: I agree with your comments. There were differences in vasculature and distribution of CD31-positive endothelial cells between day 4 and 14. We replaced the day 14 pictures to clarify this point. For your reference distribution of CD31-positive endothelial cells on day 4, 7, and 14 is shown below. These figures suggest the angiogenic process of tumor and construction of vascular network consistent with that in ASPS at the later stage, i.e., day 14.

Finally, the ChIP-seq in 2d fits with the suggestions in Fig1 and SuppFig1 that the knock-down was very poor of AT3. Really, a stronger knock-down should be achieved before trying to do ChIP-seq.

Response: Although some AT3 signals remained in the ChIP-seq analysis in Figure 2 as indicated by the reviewer, we provided the most efficient result of multiple knockdown experiments for AT3 in human ASPS cells.

My recommendation would be to expand the explanation of the CRISPR-KRAB screen and begin with that, as opposed to all the efforts to prove that AT3 loss, which is very poorly controlled in your experiments, has no impact on proliferation and only on angiogenesis.

Response: Thank you for your comment. We do not agree with the comment, and we regard the AT3 function in vivo identified in cell culture and transplantation experiments as the significant and novel finding in our manuscript.

Reviewers' Comments:

Reviewer #4:

Remarks to the Author:

Thank you for addressing most of my comments.